# A State-of-the-Art of Metal-Organic Frameworks for Chromium Photoreduction vs. Photocatalytic Water Remediation

**DOI:** 10.3390/nano12234263

**Published:** 2022-11-30

**Authors:** Andreina García, Bárbara Rodríguez, Maibelin Rosales, Yurieth M. Quintero, Paula G. Saiz, Ander Reizabal, Stefan Wuttke, Leire Celaya-Azcoaga, Ainara Valverde, Roberto Fernández de Luis

**Affiliations:** 1Advanced Mining Technology Center (AMTC), Universidad de Chile, Avenida Beauchef 850, Santiago 8370451, Chile; maibelin.rosales@amtc.cl (M.R.); yurieth.mar@gmail.com (Y.M.Q.); 2Mining Engineering Department, Faculty of Physical and Mathematical Sciences (FCFM), Universidad de Chile, Av. Tupper 2069, Santiago 8370451, Chile; 3Centro de Investigación en Recursos Naturales y Sustentabilidad (CIRENYS), Universidad Bernardo O’Higgins, Avenida Viel 1497, Santiago 8320000, Chile; elidesmar@gmail.com; 4Basque Center for Materials, Applications and Nanostructures, UPV/EHU Science Park, 48940 Leioa, Spain; paula.gonzalez@bcmaterials.net (P.G.S.); ander.reizabal@bcmaterials.net (A.R.); stefan.wuttke@bcmaterials.net (S.W.); leire.celaya@bcmaterials.net (L.C.-A.); ainara.valverde@bcmaterials.net (A.V.); 5Department of Organic and Inorganic Chemistry, Faculty of Science and Technology, University of the Basque Country (UPV/EHU), Barrio Sarriena s/n, 48940 Leioa, Spain; 6IKERBASQUE, Basque Foundation for Science, 48009 Bilbao, Spain; 7Macromolecular Chemistry Group (LABQUIMAC), Department of Physical Chemistry, Faculty of Science and Technology, University of the Basque Country (UPV/EHU), Barrio Sarriena s/n, 48940 Leioa, Spain

**Keywords:** Metal-Organic frameworks, photocatalysis, hexavalent, chromium, adsorption, water remediation

## Abstract

Hexavalent chromium (Cr(VI)) is a highly mobile cancerogenic and teratogenic heavy metal ion. Among the varied technologies applied today to address chromium water pollution, photocatalysis offers a rapid reduction of Cr(VI) to the less toxic Cr(III). In contrast to classic photocatalysts, Metal-Organic frameworks (MOFs) are porous semiconductors that can couple the Cr(VI) to Cr(III) photoreduction to the chromium species immobilization. In this minireview, we wish to discuss and analyze the state-of-the-art of MOFs for Cr(VI) detoxification and contextualizing it to the most recent advances and strategies of MOFs for photocatalysis purposes. The minireview has been structured in three sections: (i) a detailed discussion of the specific experimental techniques employed to characterize MOF photocatalysts, (ii) a description and identification of the key characteristics of MOFs for Cr(VI) photoreduction, and (iii) an outlook and perspective section in order to identify future trends.

## 1. Introduction

Hexavalent chromium (Cr(VI)) is a highly toxic metal that is stabilized as chromate oxyanions in water (Figure 1). It induces well-known cancerogenic and teratogenic effects in living organisms due to its oxidative nature. In addition, the environmental, ecotoxicology and health impacts of Cr(VI) are intensified due to the industrial wastewater effluents derived from diverse manufacturing processes such as leather tanning, cooling tower blowdown, plating, electroplating, anodizing baths’ rinse waters, etc. [1,2,3].

Depending on the pH and redox potential of the water, chromium ions can be stabilized in its hexavalent and trivalent forms, as visualized in the Pourbaix diagram (Figure 1). In parallel, the chromium oxidation state, along with the acidity/basicity of the media (i.e., pH), also governs the chromium speciation in water. Whilst trivalent chromium is usually stabilized as cationic oxo-aquo species with octahedral environments, its hexavalent form is usually found as neutral or negative oxyanions as tetrahedral H_2_CrO_4_, [HCrO_4_]^−^, [CrO_4_]^2n−^ and [Cr_2_O_7_]^2n−^.

The redox properties of Cr(VI) ions enable applying different strategies for its removal through adsorption and photo-, chemo- or electroreduction [4,5,6]. Specifically, the most explored chromium removal technologies are precipitation–coagulation, ion exchange, membrane separation, adsorption, and reduction [7,8,9,10,11]. Among them, photocatalysis offers the possibility to rapidly reduce Cr(VI) to Cr(III) without the addition or production of any hazardous chemical by-product. Therefore, water remediation systems have to be able to capture and transform highly hazardous Cr(VI) into much less toxic Cr(III) species avoiding the release of chemical to the media [12].

### 1.1. Photocatalysis for Hexavalent Chromium Detoxification

Photocatalysis is based on the generation of electron–hole pairs when a semiconductor is irradiated with a light source with an energy higher than its optical band gap (Figure 2a). The photogenerated charge carriers diffuse through the solid crystalline structure (Figure 2b), where they actively participate in varied redox reactions for the formation of reactive oxygen species (ROSs) (Figure 2c) [13,14].

The main aspects that describe a photocatalyst’s efficiency are (i) its capacity to harvest light (optical band gap) (Figure 2a), (ii) the efficiency to generate electron–hole pairs during illumination, and transport them through the crystal lattice preventing their recombination (photoconduction) (Figure 2b), and (iii) the easy generation of ROSs at the photocatalyst–water interphase, which in the end, are the ones that drive the oxidative and reductive degradation or transformation of pollutants (Figure 2c). In this respect, photogenerated electron–hole pairs, and as a consequence, some of the oxygen reactive species generated from them at the interphase between the photocatalysts and the media, have enough reduction potential to transform Cr(VI) into Cr(III) (Ev of Cr(VI)/Cr(III) redox pair is of + 0.51 V vs. NHE at pH 6.8) [15].

In parallel, the oxidative or reductive power of the hydroxyl (^•^OH), superoxide anion (O_2_^•−^), hydrogen peroxide (H_2_O_2_), singlet oxygen (^1^O_2_) or electron radicals generated at the photocatalyst–aqueous interphase are also responsible of the degradation or transformation of inorganic and organic chemicals [16]. For instance, in photocatalytic reactions for environmental remediation, numerous oxidizable compounds such as common organic pollutants or trivalent arsenic species are known [17,18,19,20,21,22,23,24,25,26]. In contrast, the reducible compounds are limited to alkyl halides, halogen oxoacids or the case of hexavalent chromium explored in this review [27,28,29,30,31,32]. Therefore, the efficiency and selectivity over the formation of ROS is a key feature for photocatalyst evaluation [33]. Although the ROS quantification protocols are well established for inorganic photocatalysts, they have been rarely applied to MOFs [34,35]. For instance, it is foreseen that there will be some re-adaptation of these protocols when applied for certain MOFs, since due to their porous nature, the adsorption of model molecules could be significantly higher than in their parent inorganic photocatalysts. In that respect, the readers may consult the deep analysis developed by Y. Nosaka and Y. Nosaka to identify and quantify the generation of reactive oxygen species by different experimental means [16].

As illustrated in Figure 2c,d, both oxygen reduction and oxidation reactions take place concurrently during photocatalysis. As a starting point of the pollutant’s degradation, a plethora of ROS with varied redox potentials are sequentially generated during illumination at the interphase between the photocatalysts and the surrounding aqueous media. The redox potentials are dependent on the pH, but also on the degree of the stabilization energy when they are adsorbed at the surface or inner pore space. Therefore, the chemical nature of the photocatalysts plays a pivotal role in generating certain ROS, as well as modulating their oxidative/reductive power when stabilized as adsorbed species at the surface of the inner pore structure of the photocatalysts.

Considering photocatalysts with ideal characteristics for water remediation purposes, to date, the most efficient class of materials are metal oxide semiconductors [34,35,36,37], chalcogenides [38,39] and carbon nitrides [40]. Overall, the current state-of-the-art challenges of photocatalysts for water remediation purposes are (i) widening the band gap to make the semiconductors functional under visible-light illumination, (ii) circumventing the carrier recombination while boosting their separation, and (iii) increase their mobility through the photocatalyst’s crystalline framework [41,42]. In this respect, doping and heterojunction engineering have gained strong attention during recent years in order to improve the carrier’s mobility [43,44,45]. Another important aspect is to endow the material with specific surface chemistry able to retain the photo-transformed intermediate and final species [46]. Consequently, the intrinsic non-porous nature of inorganic photocatalysts has strongly limited their ability to efficiently capture both the substrates and their oxidized or reduced products of the photocatalytic reaction. This bottleneck is especially important in applying a photocatalyst for the Cr(VI) detoxification of water. For instance, when working at acidic conditions, its well known that the photo-transformed Cr(III) species remains soluble. Nevertheless, the evolution of Cr(III) concentration in solution is rarely monitored in photocatalytic experiments carried out at acidic conditions [47].

### 1.2. Photocatalytic Materials for Hexavalent Chromium Reduction

Until today, a myriad of inorganic and organic photocatalytic materials have been explored for the photoreduction of hexavalent chromium. These can be grouped mainly into metal oxide semiconductors [48,49,50,51,52], chalcogenides [53,54,55,56,57,58,59,60,61,62,63,64,65,66,67,68,69], and carbon and nitride-based materials [70,71,72,73,74,75].

Metal oxide semiconductors, such as ZnO, TiO_2′_, CuO, WO_3_, NiO, NaTaO_3_, SnO_2_, CeO_2_ and BiVO_4_, have been deeply investigated for photocatalysis, including the photoreduction of Cr(VI) [48,49,50,51,52]. Overall, these materials display several practical benefits such as high generation of reactive electron/hole pairs, good optical properties, and hydroxyl-rich surface chemistry, which tends to improve the ROS generation. In addition, their chemical stability makes them robust candidates for their long-term application. In contrasts, classic semiconducting materials show a faster recombination of electron/hole pairs, low surface area, and wide band gap. Multiple strategies have been explored to limit these drawbacks. Photosensitization, doping, heterojunction construction of narrow-band gap materials, design of plasmonic metal/semiconductor systems, fine control of morphological features at the nanoscale, crystalline phase engineering, or surface chemical encoding have been explored to extend their spectral absorption to the visible region of the solar spectrum [53,54,55,56,57]. The combination of semiconductor materials to design heterojunctions (i.e., CuO/ZnO, ZrO_2_/Fe_3_O_4_, WO_3_/TiO_2_, ZnO–TiO_2_, TiO_2_–Fe_3_O_4_, TiO_2_–Cu_2_O, NiO–TiO_2_, La_2_CuO_4_/SnO_2_ [58,59,60,61,62,63,64,65,66,67,68,69]) able to outperform the individual components has been one of the most widely studied approaches to improve the photoreduction capacity of these classic materials. This improvement comes from a best harvesting of visible light and the decreasing of photogenerated electron–hole recombination, ensuring stronger redox capacity.

In parallel to metal oxides, chalcogenides have been also evaluated for the photocatalytic reduction of aqueous Cr(VI). Binary to multi-chalcogenides (i.e., MoS_2_, SnS_2_, CdS) and chalcogenide-based heterostructures (i.e., CdS@TiO_2_, SnS_2_/TiO_2_ [70], Ag–Ag_2_S/TiO_2_, SnS_2_/rGO [71] CuS/RGO, CdS/Gd_2_O_3_, ZnIn_2_S_4_/MoS_2_) have shown interesting performances for the detoxification of hexavalent chromium from water. In comparison to metal oxides, chalcogenides exhibit narrower-band gaps mainly attributed to their higher conduction and valence band positions, although less efficient to produce hydroxyl radicals [72], this makes them more efficient for absorbing in the visible light range favoring their use under sunlight. However, their higher valence position becomes them inefficient semiconductors to produce hydroxyl radicals.

Finally, carbon nanomaterials have also been used as catalysts for Cr (VI) photoreduction. Derivatives of graphene materials [73] and graphitic carbon nitride (g-C_3_N_4_) [74] stand out as one the most studied families. Particularly, some properties of graphene oxide (GO) materials such as high surface area, low thickness of 2D nanosheets and excellent electron transfer capability have allowed achieving good efficiency to Cr (VI) photocatalytic reduction [75]. Both pristine m and reduced GO (r-GO) sheets can catalyze the chromium photoreduction under sunlight/visible light. It is important to note that the performance of r-GO is lower than GO due to the intense transition to the semi-metal state of r-GO and low density of the π electron in its surface [76]. On the other hand, hybrid materials synthesized from GO and different semiconductors such as bismuth [77,78,79], metal oxides (ZnO [80,81,82], TiO_2_ [83,84,85,86,87], WO_3_ [88,89], g-Fe_3_O_4_ [90]) carbon nitride [91], quantum dots [92,93], metal sulphide [94,95,96,97,98], silver chromate [99], and metallophthalocyanines [100,101] have shown even better photocatalytic efficiency to chromium reduction in comparison with the pristine GO. The aforementioned behavior is attributed to the excellent conductivity of graphene which allows a quick and easy movement of the photogenerated electrons between hybrid materials, avoiding the recombination and consequently favoring the chromium reduction. Additionally, it has been found that increasing the porosity of GO by r-GO incorporation in those hybrid materials produced improved catalytic system performance due to a better active site distribution. The main disadvantage of GO composite catalysts found in chromium photoreduction is the industrial-scale production and reuse of these proposed catalytic systems.

In parallel, g-C_3_N_4_ has shown even better photoreduction efficiency for Cr(VI) due to its narrow band gap (∼2.7 eV) when applied under visible light [100,101,102]. Similar to the metal oxide semiconductors, g-C_3_N_4_ have been improved by its surface and interface engineering [74]. Recently, oxygen-, carbon- and sulphur-doped g-C_3_N_4_ or complex g-C_3_N_4-_based heterojunctions Ag/Bi_4_O_7_/g-C_3_N_4_ [57] have shown an even narrower E_bg_ between 1.87 eV and 2.58 eV. As a weakness, the carbon nitride system shows low stability during photoreduction under exposure to high-intensity light irradiations.

### 1.3. Metal-Organic Frameworks as Dual Function Sorbent/Photocatalysts

One of the most appealing approaches to combine the concurrent photo-transformation and adsorption of chromium species is the development of dual-function sorbent/photocatalysts [15]. Among the porous materials that can fulfil this duality, Metal-Organic frameworks (MOFs) stand out by their intrinsic high porosity, semiconductor nature, and their high degree of structural and functional tunability [103,104,105,106]. MOFs are built up from the assembly of inorganic metal oxo-units connected into three-dimensional frameworks through organic linkers with a negative net charge (i.e., carboxylates, azolates, catechols, phosphonates…) [107,108,109,110,111,112,113,114,115,116,117,118]. By applying the principles of reticular chemistry, thousands of MOFs with varied structures, topologies and functionalities have been reported. Among their most relevant and unique characteristics, MOFs exhibit (i) high crystallinity, (ii) tunable porosity, (iii) large surface area, (iv) tailorable chemistry, and (v) interfacial charge transfer properties that award them the character of a porous semiconductor under light irradiation (Figure 3) [119,120].

More specifically, the photocatalytic functionalities of MOFs arise from their versatility at compositional, chemical and porous structural levels (Figure 3) [121,122,123]. First, MOF structures with almost all the first-row transition metals, alkaline and alkaline earth, rare earth, and even actinide materials have been obtained to date [124,125]. Today, the photocatalytic reduction of Cr(VI) to Cr(III) has been driven with MOF materials built up from divalent (i.e., Zn(II), Cu(II), Co(II), Cd(II)), trivalent (i.e., Fe(III), Cr(III), In(III)) and tetravalent (Zr(IV) and Ti(IV)) ions. It is important to highlight at this point the photo-Fenton functionality of iron-based MOFs, since they add the photogeneration of radicals arising from the oxidation/reduction of iron-metal clusters to the ones generated through the hole/electron’s separation and transport due to their semiconductor nature.

In addition, metal-exchange, trans-metalation or direct crystallization of multivariate MTV-MOFs gives access to encoding complex variances of the metal sequence within the inorganic nodes of MOFs (Figure 3) [126,127]. Needless to say, the nature and the variance of the metals in the inorganic structural subunits can modulate the overall photocatalytic properties of these porous materials. It is important to mention that this strategy has not been explored yet for chromium photoreduction, although previous investigations demonstrated the improvement in the photocatalytic activity as a consequence of the metal-sequencing process. One of the most illustrative examples is the post-synthetic Ti installation into the inorganic clusters of the (Zr)UiO-66 material [128,129,130]. The complexity of metal sequencing in MTV-MOFs can be expanded up to the frontiers where the charge neutrality and structural plasticity of the MOF structure allow. For example, sequencing of up to 10 different metals encoded in the same MOF-74 structure has been reported, and in some specific frameworks, the combination of mixed valence ions is also allowed until a certain threshold [131].

Similarly, organic linkers within MOFs play a pivotal role on their light harvesting and exciton generation and transport capacity. The versatility to decorate the chemical structure of the linkers with electron donor or withdrawing groups, or to design the linker itself as an antenna or chromophore to capture certain UV-Vis radiation, has been of paramount importance to tune the light harvesting and carrier separation and transport in MOFs (Figure 3) [132,133,134]. In the specific case of Cr(VI) photoreduction, the chemical structure and functional groups installed into the organic linkers serve to modulate the adsorption affinity of MOFs over hexavalent and trivalent chromium species, but also to tune their light-harvesting and photoconduction efficiency [135]. Again, during the last few years, it has been duly proved that the chemical variance introduced by the multivariate encoding of the organic linkers within the ordered MOF structure can lead to cooperative or coupled functionalities (Figure 3). For instance, multivariate reticular chemistry offers interesting advantages to combine electron donors or withdrawing functions able to expand the band gap to the visible range, and enhance the photoconduction of the MOFs’ three-dimensional scaffolds [136]. Although this strategy has been rarely studied for Cr(VI) photoreduction, the initial results seed the light to a promising perspective to further implement the MOF potentials in this research area.

Similarly, the pore space of MOFs can be rationally encoded with specific functional groups coming from the decoration of the inorganic and organic structural units or through the encapsulation of metal, metal-oxide or metal sulphide nanoparticles, or complex poly-oxoanionic species as wolframates, molybdates or vanadate-based units (Figure 3) [137]. Although less explored for chromium photoreduction, the defect chemistry of MOFs, as well as their porosity metrics, has a great impact on the Cr(VI) adsorption capacity and kinetics. In addition, the local chemistry of linker-defective positions at the clusters can endow the material with Lewis and/or Brønsted sites. The coordination environments of these catalytic centers can be systematically varied in order to tune the catalytic activity.

Overall, the versatility of reticular materials offers multiple ways to tune the light harvesting, charge mobility and transfer. Furthermore, the ability to generate reactive oxygen species of MOF photocatalysts opens the room to coupling the oxidative processes coming from ROS species with the adsorption of the substrates and the products of the photocatalytic process in a pore space specifically designed for this function.

In parallel to the functionalization of the MOF scaffold, the development of heterojunction-structured photocatalysts has been another main cornerstone to improve the efficiency of MOF-based photocatalysts for Cr(VI) light-driven reduction. The combination of MOF materials with inorganic or organic semiconductors/conductors into unique nanostructures expands the diversity of possible functions (adsorption, photoreduction, sensing, signaling…). The dissimilarity of the electronic structures of MOFs and inorganic/organic photocatalysts allows engineered heterojunctions able to guide the electrons’ and holes’ separation and transport them during illumination [138,139].

Figure 4 summarizes different strategies that have been applied to engineer heterojunctions based on MOF materials. It is important to note at this point that not all the possibilities shown in Figure 4 have been applied for Cr(VI) photoreduction.

Generally speaking, two main routes have been established to achieve a heterostructured system: (i) semiconductor/conductor into the MOF’s pore space, and (ii) heterostructured materials based on core–shell, [140] core–antenna, 2D-assembly and 2D-supported arrays [141,142]. If confined species are considered, the encapsulation of 0D inorganic or organic nanoparticles or complexes has been widely applied for catalysis and photocatalysis purposes in general [143], but so far have not been deeply investigated for Cr(VI) photoreduction. Recently, Z. Jiang et al. performed a controlled filling of the MIL-100 pores with a TiO_2_ semiconductor, a milestone that opened the avenue to use a MOF’s pores as the mould to shape complex three dimensionally connected sub-nanometer structures [144]. This impressive example overcomes by far the catalytic performance to reduce CO_2_ of the mixture of TiO_2_ and MIL-100 components. Another elegant example is the polymerization of the monomeric unit of electronic conductive polymers as PANI that has led to composite materials able to harvest light and separate electron and hole pairs with outstanding efficiencies [145]. Regarding the heterojunctions derived from MOF-based heterostructures, besides the main example shown in Figure 4, the complexity of these systems has steadily increased, and today, patterned or 3D printed heterostructures are accessible to be obtained at the lab scale [146,147,148,149,150,151,152]. In addition, the 2D and 3D structuration of the MOFs can open the avenue to endow the systems of photonic properties that could be beneficial to further boost the photocatalytic process and coupled it to an optical sensing of the target pollutant.

After reflecting the current state-of-the-art of MOF photocatalysts in general and specific for Cr(VI) reduction, it is clear that current investigations mainly focus on the modification of the inorganic clusters and/or organic linkers. Most of the applied functionalization strategies for chromium photoreduction are based on one functional aspect. However, recent investigations point out that site coupling or site cooperation functions can be achieved by multivariate encoding the framework and the pore space in order to reach multicomponent MOF systems for synergistic catalysis. The research of MOFs for Cr(VI) photoreduction is positioned a step back in comparison to reticular materials applied for gas adsorption or drug release, or even to other photocatalytic-related applications, such as hydrogen generation, water splitting, or organic pollutant degradation. For instance, during the last few years, the digital reticular chemistry concept has been expanding rapidly to design complex tailored functionalities within porous frameworks [153,154].

After introducing photocatalysis for hexavalent chromium detoxification and MOFs for the chromium photoreduction, we wish to discuss and analyze the efficient capture and photo-transformation of Cr(VI) and Cr(III) ions with MOFs in three different sections: (i) experimental procedures to determine the adsorption and photoreduction capacity of MOFs over Cr(VI); (ii) describing the photocatalytic performance of MOFs’ build up from divalent, trivalent and tetravalent metal ions; and (iii) giving a future perspective of MOFs for chromium photoreduction. Although some reviews have been previously published within this topic [155,156,157], our review covers the most recent significant advances in the functionalization of MOFs for chromium photoreduction purposes, including a deep understanding of the mechanisms that lead to an efficient capture and photo-transformation of Cr(VI) and Cr(III) ions. Importantly, our minireview includes a comparison between the-state-of-the-art designs of MOF-based photocatalysts and MOF catalysts applied to other ends, and their possible adaptation to the specific case of Cr(VI) photoreduction is discussed.

## 2. Experimental Protocols

In this section, we aim to highlight specific experimental protocols that are key to assessing: (i) the chemical and hydrolytic stability of MOF photocatalysts, (ii) their photochemical characterization, (iii) their adsorptive and photocatalytic performance, and finally, (iv) their post-operation characterization.

### 2.1. Chemical Stability of MOFs

Intensive research has been conducted in order to determine the general rules governing the hydrolytic and chemical stability of MOFs. We advise reading the reviews published by B. Liu et al. and N. C. Burtch et al. to gain a deep insight in that respect [158,159]. As a general rule, the chemical stability of MOF materials is directly related to the chemical strength of the metal–ligand bond, which in turn is ascribed to the “hard and soft (Lewis) acids and bases (HSAB)” general principle. “Hard” applies to species which are small, have high charge states and are weakly polarizable. “Soft” refers to species which are big, have low charge states, and are strongly polarizable. That is, the strength of the metal–ligand bond in the MOF will depend on the acidity/basicity strength/softness of the acid–metal and of the base–ligand. More specifically, soft acids react faster and form stronger bonds with soft bases, whereas hard acids react faster and form stronger bonds with hard bases. Nevertheless, other factors far from the HSAB principle need to be considered as well, since the nuclearity of the inorganic clusters, their connectivity through the organic linkers, and the functions encoded within the organic linkers themselves have a great influence on the chemical strength of the MOFs [160].

Chemical and hydrolytic stability is one of the most important requirements to consider when applying MOFs as photocatalysts in aqueous environments. A progressive degradation of the MOF can lead to the loss of its efficiency, but in parallel, to the leakage of the metals and organic linkers that form their structure to the water media. This is not a minor issue when the environmental hazards of the organic molecules and the metal ions forming the MOFs are taken into account [161,162].

Assessing the chemical stability of MOFs is also a delicate task that requires the use of complementary characterization techniques (Figure 5). Most of the research studies to date are based on the chemical stability assessment of X-ray diffraction. This is the first step to assess if the crystallinity, and hence the long-range order of the crystalline structure, is maintained (Figure 5a) or partially (Figure 5a(a.3)) or fully (Figure 5a(a.4)) lost when the MOF material is exposed to the working conditions. Most of the studies have considered that if the MOF material maintains its XRD signature after the functioning period, its stability is proven (Figure 5a(a.1,a.2)). Nevertheless, it is important to note that this is an oversimplified assumption, since N_2_ adsorption measurements confirm that some MOFs that maintain their crystallinity exhibit a significant loss of porosity when immersed in water (Figure 5a,b). Therefore, the assessment of the surface is the second characterization checkpoint to assess whether the internal pore structure of the materials is maintained or has been partially or completely disrupted during operation (Figure 5a,b). In parallel, a partial dissolution during the adsorption or photocatalysis experiments could lead to a significant leaching of the metal and/or linkers of the MOF, without leading to the loss of the long-range order of its structure [163]. For instance, some studies have reported a significant leaching of the metal ions (quantified by inductive-coupled plasma spectroscopy) and of the organic linkers (quantified by the UV-Vis) when the MOFs are immersed in a water solution of different acidity/basicity. Therefore, a complete description of the chemical and hydrolytic stability of the MOFs requires a third characterization step where the concentration of the metals or organic linkers in the operation media is monitored to quantify the percentage of the MOF dissolved (Figure 5c). All the characterization protocols described above are shown Figure 5, where red-, orange-, green-, and blue-colored data have been used to illustrate the XRD, BET, and UV-Vis fingerprint of chemically robust, intermediate, weak, and unstable MOF materials. It is important to note that while robust and intermediate MOFs could exhibit the same XRD or BET curves after working conditions, the less robust materials could lead to the release of some of its component to the media, as illustrated by the UV-Vis spectra shown in Figure 5c.

### 2.2. Photochemical Characterization

The term “band gap” refers to the energy difference between the top and bottom of the valence and conduction bands in a semiconductor. The electronic structure of semiconductor materials determines the energy barrier between the conduction and valence bands that is related to the energy input that is necessary for an electron to jump between them. In the specific case of photocatalytic processes, the band gap determines the energy of the photons (wavelength of the light source) that is needed to trigger the exciton generation through the electron transfer from the valance to the conduction band [164,165].

When a photocatalyst has a large exciton-binding energy, it is possible for a photon to have just barely enough energy to create a bound electron–hole pair, but not enough to separate the electron and hole pairs. In this case, the optical band gap and the electrical band gap (or “transport gap”) coincide. That is the case of most inorganic semiconductors. In this case, the optical band gap is the threshold for photons to be absorbed, while the transport gap is the threshold to separate electron–hole pairs.

In the specific case of most of the MOFs, due to the mismatch between the HOMO and LUMO levels of the organic linkers and inorganic clusters, together with the strong electron localization into these struts, their electronic structure is mainly governed by the electronic structure of their separate constituents [166]. That is, significant differences are found between their optical and transport band gaps of MOFs. Thus, the experimental determination of both parameters is key to explaining the photocatalytic performance of these materials.

The optical band gap of a solid material is determined via UV-Vis spectroscopy [167,168]. Heterogeneous catalysts are basically inspected as densely packed powders to measure the scattering of photons using the Kubelka–Munk (K–M) equation, which describes the optical properties of a photocatalyst sample by using an effective scattering (*S*) and absorption (*K*) coefficient, as described in the Equations (1) and (2).

(1)
$$F\left(R_{\infty }\right)=\frac{{\left(1-R_{\infty }\right)}^{2}}{2R_{\infty }}=\frac{K}{S}$$


(2)
$$F\left(R_{\infty }\right)=\frac{\varepsilon^{C}}{S}=$$

where

(3)
$$\left(R_{\infty }\right)=\frac{\left(R_{Sample}\right)}{\left(standar\right)}$$

stands for the diffuse reflectance of dilute species determined by the Beer–Lambert law. For instance, if a semiconductor powder of an “infinity” thickness is considered, the optical band gap energy can be calculated by plotting:
(4)
$$\alpha^{n}=vs.\hbar \omega$$

where *n* value depends on the indirect (i.e., ½) or direct (i.e., 2) nature of the band gap. More specifically, the Tauc band gap model and the Tauc plots can be described on the basis of Equations (5) and (6).

(5)
$$\omega_{2}\in_{2}\;\approx {\left(\hbar \omega \;-\;E_{g}\right)}_{2}$$


(6)
$$\alpha \left(\hbar \omega \right)\;\approx \left(\hbar \omega \;-{\;E}_{g}\right)$$


The determination of the optical band gap is the first characterization step to understand the optical properties of a photocatalyst, information that afterwards will aid adapting the source of illumination (i.e., wavelength equal or below to the optical band gap energy) during the photocatalytic experiments. As an illustrative example, the Tauc plot and the band gap calculation for two hypothetical MOFs are plotted Figure 6a. The curves are plotted with blue and green colors to illustrate how the E_bg_ of the blue-colored curve is correlated with materials able to adsorb a wider energy of light wavelengths than the one illustrated with a green color.

On the other hand, X-ray photoelectron spectroscopy (XPS) and ultraviolet photoelectron spectroscopy (UPS) techniques can be used to determine the energy of the valence band (VB) [169,170,171]. From XPS spectra, it is possible to measure the values of the valence band maximum (VBM)-binding energy. From the VB spectra, it is possible to linearly fit the leading edge of the VB and the flat energy distribution to the VB spectrum, the intersection of these two lines allows finding the VBM value, whilst the UPS spectrum is employed to determine the width of binding energy (ΔE) but gives the possibility to estimate, in an indirect way, the VB. Herein, the width value of He is used as the standard. After determining the ΔE of the studied material, the width of He (21.22 eV) is subtracted for calculating the VB value.

Before the band gap characterization, Mott–Schottky and photocurrent experiments give access to understanding the nature of the semiconducting process, and its efficiency to separate and transport the electron and hole pairs. To this end, the photocatalyst is usually integrated into an electron-conductive transparent electrode (usually an indium-tin-oxide-coated glass slide). The system is connected in a three-electrode configuration (a working electrode, Ag/AgCl reference electrode, and Pt counter electrode) while immersed in a liquid electrolyte. For n-type semiconductors, Mott–Schottky plots exhibit a positive and linear slope which is related to the flat-band potential versus the reference electrode. The flat-band potential of the n-type semiconductor (intercept value at the *x*-axis) can be used to estimate the conduction band (CB) of the semiconductor. By comparing the potential of the conduction band, it can be estimated to what extent it is more negative than the redox potential for a given reaction, such as the generation of radical oxygen species through the oxygen reduction (e.g., O_2_ → · O_2_^•−^, −0.13 V vs. Ag/AgCl) or the Cr(VI) to Cr(III) transformation (+1.15 V vs. Ag/AgCl) [172,173]. Just to illustrate one of the possible scenarios, UiO-66-NH_2_ is one of the most studied MOFs for chromium photoreduction, and the potential of its conduction band has been estimated to be close to −0.5 V vs. Ag/AgCl. This potential is low enough to trigger the direct O_2_ → O_2_^•−^ reduction, but it is not negative enough to face the two-electron reduction of water into hydroxyl radicals at basic conditions (i.e., −0.828 V vs. Ag/AgCl at pH 11.7—see Figure 2d). Usually, as illustrated in Figure 6b, the slope of the Mott–Schottky curve is related to the data obtained from the optical band gap, and in parallel, with the photocurrent experiments of our model materials (Figure 6c).

Photocurrent experiments are performed with the same three-electrode configuration used to measure the Mott–Schottky curves, but instead of scanning the variation in the capacitance vs. the potential, the photocurrent response of the material in dark and illumination conditions is measured in an open-circuit configuration mode. The absolute value of the photoconduction (always normalized to the area of the working electrode) is related to the capacity of the material to generate excitons and separate their electron and hole components efficiently. In parallel, the profile of each single pulse (i.e., dark–illumination–dark cycle) in photocurrent experiments also offers information about how fast the material is able to respond to the illumination to generate and transport the hole/electron pairs (Figure 6a,c).

Finally, the efficiency of the ROS to oxidize/reduce chemicals is also modulated by their stabilization/coordination at the surface of the photocatalyst (Figure 6d). As an illustrative example, TiO_2_-anatase semiconductor materials crystallized as nanoparticles and nanotubes with different sizes, morphologies, and crystal facets show significantly different capacities to generate ROS species during illumination. In the model MOF materials used to illustrate the photochemical characterization strategy described in this review, even though the band gap and electroconduction of the MOF illustrated by the blue-colored curves of Figure 6a–c are better than the ones of the MOF model of the green curves, the generation of ROS species could be not directly related to these factors, as shown in Figure 6d. As mentioned before, although ROS quantification protocols have long been established for inorganic photocatalysts, their application to Metal-Organic framework materials is still scarce [174,175]. As a general rule, most of the applied protocols to indirectly determine the ROS generation are the addition of oxygen radicals or electron or hole scavengers. Thus, their effects on the photocatalytic reaction give an indirect clue of the radicals involved in the process. Just to mention one of the multiple examples reported in the bibliography, *p*-Benzoquinone (PBQ), sodium oxalate (Na_2_C_2_O_4_), *tert*-butyl alcohol (TBA), 2,2,6,6-tetramethylpiperidine (TEMP), and sodium iodate (NaIO_3_) can be used as O_2_^•−^, holes, ^•^OH, [1] O_2_, and electron scavengers in the solution [176,177].

Electron paramagnetic resonance (EPR) spectroscopy performed in specific spin trap molecules has been applied to qualitatively detect the ^•^OH, ^1^O_2_, and O_2_^•−^ radicals generated by MOF photocatalysts [178]. The quantification of the radical generation during illumination conditions has been mainly reported for MOFs applied for photodynamic therapies [179,180] but it is starting to be applied for MOFs that are used for environmental purposes. Quantification of ROS is performed by colorimetric and fluorescence probe methodologies, where the concentration of the ROS is linked to the absorbance/fluorescence gain or loss or specific probe molecules. When selecting the protocol, it is important to consider the selectivity of the probe chromophore or luminescence probe over the target ROS, as well as their potential adsorption in the MOF, and their time and chemical stability.

To gain a broader perspective of the ROS detection and quantification protocols, we advise reading the review of Nosaka et al. [16]. It is important to note that ROS detection has been largely overridden in chromium photoreduction studies performed with MOFs. Due to the fact that the Cr(VI) to Cr(III) reaction is electron-consuming, the efficiency of a photocatalyst to reduce Cr(VI) is usually associated with its capacity to donate electrons to the Cr(VI) oxyanions. Nevertheless, the ROS generation involves oxidative paths that can serve as a secondary electron source that can participate in side reactions during the Cr(VI) photoreduction process [181].

### 2.3. Adsorption Kinetics/Capacity and Photocatalysis

Before performing the Cr(VI) to Cr(III) photoreduction experiments, it is important to evaluate the adsorption kinetics and capacity of the MOF photocatalysts to uptake Cr(VI) and Cr(III) ions in dark conditions. This initial characterization step will help to identify the experimental conditions of the photocatalysis experiments (i.e., initial chromium concentration, MOF loading, adsorption time in dark conditions…) (Figure 7).

In addition, the development of batch experiments to determine the adsorption capacity and kinetics of MOFs over Cr(VI) and Cr(III) ions give highly valuable information to understand to what extent these dual sorbent/photocatalysts are able to capture these species during the Cr(VI) to Cr(III) photoreduction process (Figure 7a,b) [182,183]. It is important to keep in mind that Cr(III) ions are only soluble at acidic conditions, so adsorption needs to be assessed at pHs < 4 (this acidity value also depends on the Cr(III) concentration). Adsorption isotherms are obtained from batch experiments where a known amount of MOF photocatalyst is immersed in model single-element chromium solutions of increasing concentrations. In parallel, the kinetics of adsorption are obtained by monitoring the time evolution of the chromium concentration during the adsorption process (Figure 7b). Usually, MOFs show a rapid one-step adsorption over metal ions, capturing most of the species from the solution during the initial stage of the process (below 30 min). Afterwards, the adsorption kinetics is gradually slowed down until equilibrium is reached (usually ~2–4 h). Both experimental data (kinetics and adsorption isotherms) can be fitted on the basis of different models (i.e., Langmuir, Freundlich, pseudo-first- and pseudo-second-order kinetics…) that parametrize the adsorbing capacity and affinity of the sorbent for the substrate [184,185,186,187,188]. Overall, the isotherms’ fitting gives access to quantifying the maximum adsorption capacity, as well as the adsorbate–adsorbent affinity of the MOF over a specific substrate. In parallel, the fitting obtained from kinetic models allows quantifying the rate of the process. In Figure 7a,b, two kinetic and isotherm curves for the adsorption of Cr(VI) and Cr(III) species by two model MOFs are illustrated. The blue-colored curves show the response of MOF materials with a high affinity to capture Cr(VI) species and a negligible capacity to adsorb Cr(III), while the green-colored curves illustrate the response of a MOF with intermediate affinities and capacity to capture both species. The surface area, pore window aperture, and presence of preferential adsorption points within the MOF structure are some of the characteristics that will shape their kinetic and isotherm adsorption profiles. That is, both the adsorption capacity and affinity of the MOFs over chromium will be shaped by the fast diffusion paths across the framework, and by the density and chemico-physical affinity of preferential adsorption sites over chromate oxyanions or trivalent chromium cationic species.

After the equilibrium during the adsorption stage is reached, the photocatalysis process is triggered by illumination. It is well known that the wavelength energy of the light source is key to enhancing the electron and hole separation within the semiconductor materials. For instance, the energy input needs to be higher than the band gap of the semiconductor to trigger the process. Nowadays, the photocatalytic reactors allow tuning easily the source of light used to perform the experiments. For instance, a broad scope of lamps, light-emitting diodes (LEDs), and laser sources are available to select the range of wavelengths (i.e., lamps and LEDs) or the specific wavelength (i.e., laser) to perform the experiment. It is important to mention that performing the photocatalysis experiments under different light sources opens the perspective to understand the photocatalyst’s performance, but from the application perspective for water remediation purposes, the use of a light source with a spectral fingerprint close to the one of the sunlight is highly desired [189,190,191].

Besides illumination source, there are varied but important parameters that have a great influence on the Cr(VI) to Cr(III) photoreduction process: the loading of the catalysts (i.e., with a usual operative window between 0.25 and 1.0 g·L^−1^); the initial Cr(VI) concentration after reaching the adsorption equilibrium in dark conditions; the acidity/basicity of the aqueous media; or the presence of competitor species as chloride, sulphate, or carbonate anions. The latter three parameters, and especially the pH, affect the hexavalent chromium speciation in solution, and hence the affinity of the MOFs to adsorb and photo-transform them into trivalent chromium. MOF materials with negative surface charge could electrostatically repel the Cr_2_O_7_^2−^ anions, avoiding their adsorption. In contrast, when the pH value is below the isoelectric point of the photocatalyst, the zeta potential of this is positive; therefore, the positively charged surface of the catalyst can be expected to provide better adsorption performance for Cr_2_O_7_^2−^ anions, and in parallel, the best photocatalytic behavior is observed (Figure 7c,d).

All the parameters that affect the photocatalysis also have a correlative influence during the oxidative and reductive radicals’ generation by the MOF semiconductors. Although many experimental procedures have been reported for inorganic photocatalysts to directly or indirectly measure the radicals generated during illumination, their application to MOF materials is still limited.

Last but not least, the quantification of the chromium concentration during the adsorption and photocatalysis experiments is also a key aspect. The concentration of hexavalent chromium can be determined by means of the diphenyl carbazide protocol, via the absorbance at a specific wavelength of the UV-Vis spectra of the chromate solution, or by inductively coupled plasma spectroscopy. The latter one is the most used experimental procedure to determine the overall chromium concentration (i.e., Cr(VI) + Cr(III)) before, during, and after the adsorption and photocatalysis experiments. For instance, during adsorption, no change on the oxidation state of the chromium ions in the aqueous solution is expected, although recent findings point out that specific MOF functionalities can lead to a chemical reduction of the Cr(VI) to Cr(III) once the ions are immobilized within the pore structure of these materials. To measure the concentration of Cr(VI) and Cr(III) ions is another important aspect in order to unravel if the photoreduced Cr(III) ions are stabilized within the dual sorbent/photocatalyst or are leached to the media after their transformation (Figure 7a,c,d). It is necessary to determine the Cr(VI) concentration by means of the diphenyl carbazide protocol, and in parallel, the overall chromium concentration through ICP-AES analyses. The difference between these two values gives access to determining the evolution of the Cr(III) content into the aqueous solution during the photocatalysis experiment.

Taking into account all these abovementioned factors, the two possible responses of MOF photocatalysts during chromium photoreduction are illustrated in Figure 7c,d. In Figure 7c, the MOF is able to efficiently photo-reduce the Cr(VI) to Cr(III), but due to its negligible capacity to adsorb Cr(III), this is released to the media during photocatalysis. Opposite, in Figure 7d, the response of a model MOF that is able both to photo-transform the Cr(VI) to Cr(III), and in parallel, to adsorb the photogenerated Cr(III) species, is illustrated.

### 2.4. Post-Operation Characterization

Besides the usual experimental techniques applied to characterize MOFs before and after operation (i.e., X-ray diffraction, N_2_ adsorption isotherms, IR, H-NMR, thermogravimetric analysis…), X-ray photoelectron spectroscopy (XPS), solid UV-Vis, IR and electronic paramagnetic spectroscopies (EPR), and X-ray absorption (XAS) play an important role in unraveling the chromium oxidation state, coordination environment, and bridging modes to the MOF host structure, once they are immobilized within the porous frameworks. XPS is one of the most employed techniques to study both the binding energies of the chromium ions and of the MOF structure after adsorption or photocatalytic experiments. For instance, the binding energies of Cr_2p_ are slightly different for Cr(III) and Cr(VI) ions, allowing their differentiation by a careful fitting of the spectra. This is schematically illustrated in Figure 8a, where the contribution of Cr(III) and Cr(VI) ions to the overall XPS spectra is highlighted with different colors. In addition, it is important as well to follow up the binding energies of the Zr(IV) ions, oxygen atoms of the hydroxyl groups located within the zirconium hexanuclear units, as well as any functional group (e.g., NH_2_, OH, NO_2_, SO_3_…) encoded within the organic linkers of the MOF [192]. Depending on the adsorption mechanisms, the chromium ions could alter the binding energies of these specific functions within the MOF involved in the chemisorption, adsorption, chemical reduction, or photoreduction process (Figure 8a. These subtle variations in the local functionalities of the MOFs have been followed as well by other spectroscopic characterization techniques such as IR, Raman, or Mossbauer spectroscopy [193].

The color dependence of chromium ion species on their oxidation state and coordination environment opens the opportunity to study their speciation by solid UV-Vis spectroscopy. For instance, hexavalent chromate oxyanions exhibit a characteristic orange color that gives rise to UV-Vis bands located at ~35,000 cm^−1^ (~285 nm) and 27,000 cm^−1^ (~370 nm). This signal is attributed to the Cr^6+^-O^2−^ charge transfer for mono-chromate species. When a chemical of a photoreduction process occurs within the MOF, Cr(III) ions are stabilized within the framework. As a result, the material acquires the usual green color associated with trivalent chromium species. UV-Vis spectra of the MOF after the chromium immobilization can help to identify and semi-quantify these species, since the UV-Vis spectra can exhibit the characteristic signals associated with the spin-allowed d–d transitions of an octahedral coordinated Cr^3+^d^7^, ^4^A_2g_(F) → ^4^T_1g_(F) (~23,000–24,000 cm^−1^/~435–417 nm) and ^4^A_2g_(F) → ^4^T_2g_(F) (~20,000–17,500 cm^−1^/~500–571 nm) (Figure 8b) [194]. The model UV-Vis fingerprint of Cr(VI) and Cr(III) ions, found once installed in MOFs, is depicted in Figure 8b.

In a parallel approach, EPR spectroscopy can be applied to find out the presence of trivalent chromium. Figure 8c and d illustrate the main EPR signals that can be found for the Cr(III) isolated species (δ-signal), Cr(III) clustered species (β-signal), and Cr(V) transient species (γ-signal). For instance, depending on the experimental data, the presence of isolated or clustered Cr^III^ ions can be differentiated by EPR spectroscopy (Figure 8c). Furthermore, EPR gives access to determining if pentavalent chromium species are also stabilized within the MOF as well (Figure 8a,c). For a more detailed analysis of the chromium speciation by EPR, we advise reading the work of P.G-Saiz et al. [195,196,197].

Finally, X-ray absorption is the forefront experimental procedure not only to unravel the oxidation state of the chromium ion before, during, and after operation, but the local structure when immobilized in the Metal-Organic framework porous structure. The term X-ray absorption fine structure (XAFS) is used to refer collectively to both the X-ray absorption near-edge structure (XANES) and the extended X-ray absorption fine structure (EXAFS) regions [198,199]. These two regions are differentiated on the basis of the dominant electronic processes in each region. In the XANES region, the multi-scattering of outgoing and backscattered photoelectron waves between absorber Cr atoms and surrounding atoms shapes the main adsorption edge of the XAS spectra. In contrast, the EXAFS region is dominated by the interferences between singly scattered outgoing and backscattered photoelectron waves that cause most of the oscillatory features of this high-energy region of the spectra.

Although this technique has not been applied to study MOFs for chromium adsorption or photoreductions, it has the potential to unravel the local coordination environment and structure of metal ions adsorbed or installed by post-synthetic procedures (i.e., metalation, chemical vapor deposition…). XAS has been applied in other sorbents to follow up the Cr(VI) capture and transformation by studying the near-edge structure of the X-ray absorption spectra (XANES) [200]. For hexavalent chromate anions, a prominent pre-edge peak at 5990 eV is usually observed, which arises from a bound-state 1s to 3d transition, allowed for non-centrosymmetric Cr(VI)O_4_ and forbidden for centrosymmetric Cr(III)O_6_ octahedra. The size of this pre-edge peak can be used to quantify the proportion of Cr(VI) in a sample if the Cr(VI) fraction makes up greater than ~1–5% of the total Cr present. In the case of Cr(III) ions, small pre-edge features are present for octahedral Cr(III) at 5990.5 and 5993.5 eV due to 1s to 3d(t_2g_) and ls to 3d(e_g_) electronic transitions, respectively (Figure 8a,d). The EXAFS oscillatory fingerprint of the XAS spectra can be fitted by simulating the atomic scattering amplitudes and phase shifts (FEFF) of given local structural models for chromium species. Usually, the structural information of known Cr(VI)- and Cr(III)-based compounds is used as the starting point of the fitting process. Usually, both the number of neighboring atoms (N) and their distance, Å, from the absorber Cr atom can be accurately calculated with an accuracy equal to or below 0.03 Å. In addition, for some cases, the Debye–Waller factors (i.e., static and vibrational atomic disorder) can be obtained as well from the EXAFS fitting. Figure 8e,f illustrate the model XANES spectra of Cr(VI) and Cr(III) ions (Figure 8e), and of the possible radial bond distances’ distribution obtained from the XAS data treatment (Figure 8f).

## 3. Metal-Organic Frameworks for Hexavalent Chromium Photoreduction and Capture

Here, we wish to discuss the most relevant investigations carried out with (i) divalent, (ii) trivalent, and (iii) tetravalent metal-based MOFs for chromium photoreduction. Within these three subsections, the state-of-the-art of MOF technology for chromium detoxification of waters will be comprehensive analyzed. Furthermore, we wish to highlight the milestones achieved within this area, and compare them to the most recent reticular material developments for adsorption, catalysis, and photocatalysis.

### 3.1. Divalent Metal-Based Metal-Organic Framework Photocatalysts

Considering the photocatalytic ability of a ZnO semiconductor, the use of Zn-based MOFs in the photoreduction of chromium has been a natural step of exploration as a first approach to test the feasibility of MOF photocatalysts for hexavalent chromium detoxification (Table 1).

The research on Cr(VI) photoreduction is limited to ZIF-8, BUC-21 (Zn(II)/anthracene), and NNU-36 materials and their composite structures when combined with inorganic semiconductors. In the specific case of the well-known ZIF-8 zeolitic imidazole framework (ZIF), although active for Cr(VI) photoreduction, its wide band gap (5.2 eV) severely limits its efficiency to harvest light and trigger the photocatalytic process. A band gap narrowing has been achieved by engineering Zn-MOF based on chromophore carboxyl-based organic linkers with aromatic ring systems. First, BUC-21, which is a Zn-MOF build up from Zn-paddlewheel and 1,3-dibenzyl-2-imidazolidone-4,5-dicarboxylic acid square planar “carboxylate-metal organic layers” pillared by a 4,4′-bipyridine (bpy) secondary linker, has been studied. The coordination environment of the paddlewheel units differs significantly from the one shown by the Zn(II) ions in ZIF-8, also inducing a shift in the optical band gap to 3.4 eV. The material exhibits a better Cr(VI) photoreduction response in comparison to the ZIF-8. Surprisingly, even if the long-term hydrolytic stability of Zn-MOFs is in question, BUC-21 exhibits excellent reusability [201].

**Table 1 nanomaterials-12-04263-t001:** Divalent-metal-based MOF photocatalysts for Cr(VI) to Cr(III) reduction.

Metal Center	MOFs	pH	Light Source	[Cr (VI)]_0_ (ppms)	Photocatalyst Loading (g/L)	Photo-Oxidation Efficiency		Ref.
						Removal Percentage (%)	Time (min)	
Zn	ZnO@ZIF-8	7	UV	20	1	88	240	[202]
	ZIF-8@Cd_0.5_Zn_0.5_S	6	Vis.	20	1	100	10	[203]
	MoO_3_@ZIF-8		Vis.	20	0.5	96	40	[204]
	ZIF-8@CuPd	1	Vis.	20	0.20	89	60	[205]
	BUC-21	2	UV	10	0.75	96	30	[205]
	TNT@BUC-21	5	UV	10	0.16	100	20	[206]
	BUC-21 and g-C_3_N_4_	2	SL	10	0.25	100	60	[207]
	BUC-21 and Bi_24_O_31_Br_10_	2	Vis.	10	0.25	99	120	[208]
	NNU-36	2	Vis.	10	0.38	95.3	60	[209]
	MOF-Zn-BPEA	3	Vis.	10	0.38	92	50	[210]
	Zn-MOF ^[a]^	2	SL	20	1	93	90	[206]
	MIL-101/Pd-Cu	NR	Vis.	NR	NR	100	30	[211]
	Zn-PA-MOF	2–6	UV	20	0.4	98	90	[212]
Cd	BUC-66	2	UV	10	0.075	98	30	[213,214]
Co	BUC-67					99	30	
Cd	Cd(4-Hptz)_2_.(H_2_O)_2_]_n_	3	UV	10	0.175	100	50	[210]

^[a]^ 1 mL EtOH as scavenger. Vis. = visible light, SL = sun light, WL = white light.

Representing a step forward, the incorporation of visible-light-responsive bipyridine-like linkers, such as 9,10-bis(4-pyridylethynyl)-anthracene (BPEA) acting as a pillars of 2D Zn-carboxylate layers, allows expanding the optical band gap of pillared MOFs to the visible-light region, and thus improving the photocatalytic efficiency of the material under sunlight illumination [215].

The photocatalytic activity of Zn-MOFs is not only limited to robust 3D and 3D porous structures, but to one-dimensional ladder-like Zn(II)/BPEA coordination polymers [216]. The incorporation of a chromophore ligand is a well-known and widely applied strategy to tune the visible-light-harvesting capacity of MOF and, thus, improve their Cr(VI) photoreduction efficiency. The luminescence properties of anthracene-like organic linkers have been applied as well to detect hexavalent chromium as the presence of chromate anions induces a quenching of the luminescence signal. Although scarcely explored, other divalent Cd(II) [213], Co(II) [214], and Cu(II) [217] MOFs have been also successfully tested for hexavalent chromium photoreduction purposes. In terms of band gap energy, there is a clear advantage when applying Co(II)- or Cu(II)-based MOFs for photocatalysis, since their light-harvesting capacity is shifted to the visible range. In addition, copper is a well-known active metal center for the oxidative catalytic degradation of organic pollutants such as phenols. For instance, as the oxidative degradation by copper sites depends on the generation of oxygen radicals such as hydrogen peroxide, exciton generation through light illumination can induce an enhancement in ROS generation, Cu center activation, and finally, organic pollutant oxidation [218,219,220].

The engineering of MOF-based heterojunctions has allowed tuning the optical band gap energy to visible-light capture and improving the overall properties and photocatalytic performance of Zn-MOFs. The soft synthesis conditions of divalent MOFs make the in situ growing of the MOF at the surface of different materials relatively straightforward [150]. Three are the main strategies that have been explored to construct heterostructured MOF materials for chromium photoreduction:(i)The physical mixture through ball milling of MOF and other semiconductor inorganic or carbon-based materials (i.e., Bi_24_O_31_Br_10_ nanoparticles, and graphitic carbon nitride (g-C_3_N_4_); (Figure 9a—Table 1).(ii)The direct growth of metal or sulphide nanoparticles at the surface of the ZIF-8 nanosized crystals (i.e., ZIF-8@CuPd [221] and ZIF-8@Cd_0.5_Zn_0.5_S [222]) (Figure 9b, Table 1).(iii)The generation of semiconductor–MOF core–shell structures (i.e., ZnO@ZIF-8 nanoparticles, MoO_3_/ZIF-8 nanowires, and TiO_2_@BUC-21 nanotubes (Figure 9c, Table 1).

**Figure 9 nanomaterials-12-04263-f009:**
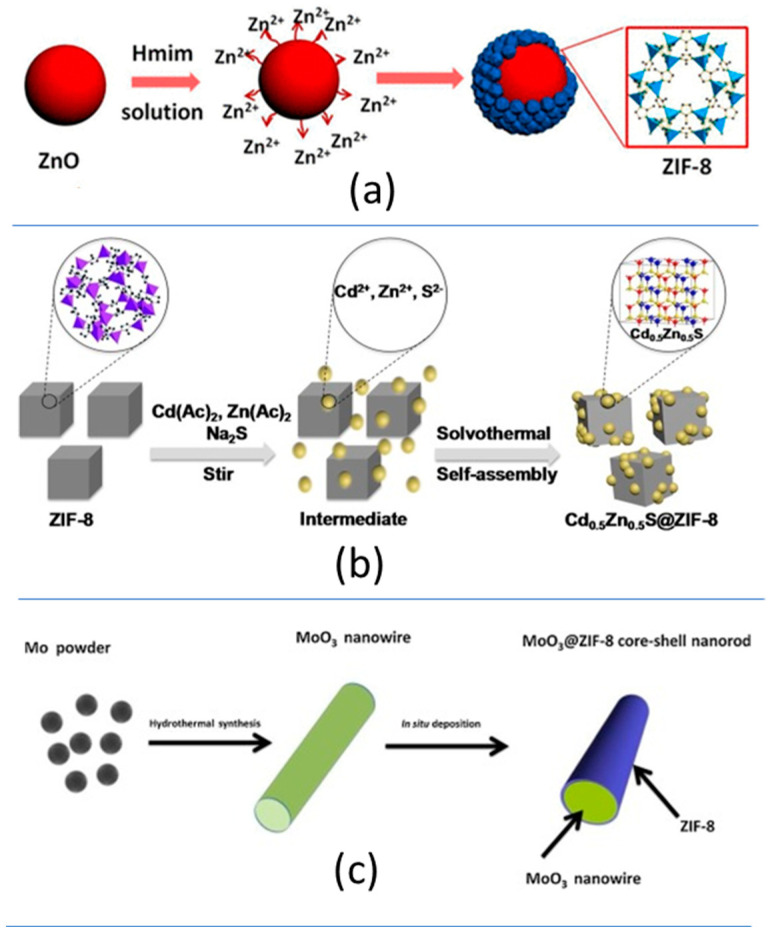
Heterostructures based on divalent metal MOFs (**a**) Core-shell ZnO@ZIF-8 nanoparticles, (**b**) ZIF-8@CdZnS core–antenna metal–sulphide nanoparticles and (**c**) MoO_3_@ZIF-8 core–shell nanowires. Figures adapted from references [125,126,127].

In general terms, heterojunctions obtained from MOFs and inorganic/organic semiconductors lead to a shift in the optical band gap energy to the visible-light energy range that improves the light-harvesting capacity of these composites in comparison to their individual components. Merging the electronic structures at the interphase between the MOF and classic semiconductor materials also leads to improved photoconduction of the composites in comparison to their parent components. For some cases, the composite materials have been revealed as multifunctional photocatalysts able to couple the chromium photoreduction to the photo-oxidation of organic pollutants as methylene blue dye (Figure 9—Table 1).

Although the chemical stability of the Zn-MOFs under the working conditions usually employed in photocatalysis have been confirmed in many works through X-ray diffraction after operation, and the recyclability of some of the Zn-MOFs indicates that a minor loss of activity is observed, it is important to be cautious when evaluating these conclusions, even when an important number of the studies have been performed at low-pH conditions. As explained before, the chemical and hydrolytic stability of Zn-carboxylate and Zn-imidazole bridges are limited, especially at acidic conditions, and the Zn-MOFs usually lose their porosity and long-range ordering when exposed to moisture conditions or immersed in water in the short- to mid-term. So, even if the materials could resist the chemico-physical conditions of the Cr(VI) to Cr(III) reactions in usual experiments carried out at lab-scale, this does not preclude the MOF being partially dissolved or disintegrated during the process. The environmental impact of a partial leaching of the MOF during the operation will depend on the toxicity of their components, which is especially concerning when sophisticated organic linkers based on aromatic rings or pyridyl moieties are used to build up the MOF structure.

### 3.2. Trivalent-Metal-Based Metal-Organic Framework Photocatalysts

Among the trivalent transition metals employed for the synthesis of MOFs, iron, chromium, and aluminum are the most investigated ones (Table 2). Depending on the synthesis conditions and the connectivity and functional groups of the organic linkers, Fe(III), Al(III), and Cr(III)-MOFs can crystallize in a wide variety of structures with high-to-intermediate hydrolytic and chemical stabilities (Figure 10).

Although Fe-MOFs exhibit lower chemical and hydrolytic stability than chromium and aluminum homologues, this family of MOFs has been the most profoundly explored in terms of their photocatalytic activity to face hexavalent chromium pollution. The cost-effectiveness, facile fabrication, environmental friendliness, and excellent photosensitivity of the Fe-MOFs make these materials highly appealing for environmental remediation purposes. Even if the chemical strength of Fe-MOFs is lower than chromium, aluminum, or zirconium homologues, the environmental risks derived from the leakage of their components to water media is null or very low if the adequate organic linkers are selected. Furthermore, in addition to the usual photocatalytic mechanisms triggered by the illumination of the iron MOFs, Fenton-like functions of iron-based materials can generate additional radical species via metal-redox-related pathways. For instance, the ROS generated via photocatalysis can activate the Fenton catalysis as well.

Among the Fe(III)-based MOFs studied for the photoreduction of Cr(VI), MIL-88B, MIL-53, MIL-101, and MIL-68 (build up from the1,4-benzenedicarboxylic acid (BDC)) and MIL-100 (assembled from trimethyl 1,3,5-benzenetricarboxylate (BTC)) are the most applied ones. Although they share common or very similar building blocks, their crystal structures differ significantly in terms of surface area, connectivity of the iron-oxo units, and connectivity of the pore space. For instance, MIL-53 and MIL-68 structures exhibit inorganic chains of corner-shared iron-oxo units with an octahedral environment that are connected through the BDC organic linkers to form a three-dimensional structure with one-dimensional pores (Figure 10).

In comparison, the crystal structures of MIL-100 and MIL-88 compounds are built up from the archetypal trimeric Fe-units, which are connected in a three-dimensional framework via the BTC and BDC organic linkers (Figure 9). For more detailed information of the topology, porosity, and properties of these materials, readers may consult reference [156]. In comparison, the crystal structure of Fe-MIL-53 is constructed from one-dimensional Fe-oxo inorganic chains. These 1D interconnected paths within MIL-53 make it a better photoconductor in comparison to MIL-100 and MIL-88. In addition, MIL-53 shows excellent performance at slightly acidic (pH 4) to highly acidic conditions (pH 1), it is functional under visible-light illumination, and its reusability gives rise to a negligible loss of its efficiency (Table 2). Among the underlying mechanisms that can explain this performance, the direct excitation of iron-oxo cluster is one of the most interesting proposals, as reported by Laurier et al. [246]. Thus, the iron-oxo chains in MIL-53 adsorb incident photons under visible-light irradiation, and the photogenerated charge carriers migrate to the surface of the MOF particles participating in the redox reaction. Afterwards, the reduction of Cr(VI) to Cr(III) absorbed on the surface is driven by the photogenerated electrons (E_CB_ = −0.40V vs. NHE at pH 6.8 and E_Cr(VI)/Cr(III)_ = + 0.51V vs. NHE, pH 6.8). Fenton mechanisms can also play an important role by directly generating (i) electrons able to potentiate the Cr(VI) photoreduction, and (ii) reactive oxygen species able to activate the iron sites to induce a Fenton parallel reaction [247]. In particular, the good functionality of Fe-MIL-68 under neutral to slightly acidic conditions (pH = 7–5), or its outstanding performance under acidic environments <5, further evidences the versatility of Fe-MOFs to function under varied environmental conditions (Table 2).

There is plenty of room for innovation in terms of organic linkers’ functionalization in Fe-MOFs applied for chromium photoreduction. To date, most of the chemical encodings are limited to the incorporation of amino groups to the organic linkers, as are the cases of MIL-53-NH_2_, MIL-88B-NH_2_, and MIL101-NH_2_ materials (Table 2).

Fe-MOFs, in comparison to their non-functionalized variants, is attributed to the dual excitation pathways achieved through (i) the photoinduced exciting of amine functionality followed by the electron transfer to the Fe3-μ_3_-oxo clusters and (ii) the direct excitation of Fe3-μ_3_-oxo clusters (Table 2). Just as an illustrating example, MIL-88B fails to complete the Cr(VI) reduction under visible-light irradiation (i.e., 20%) while MIL-88B-NH_2_ reaches the complete Cr(VI) to Cr(III) photoreduction under the same conditions. In fact, MIL-88B-NH_2_ outperforms the efficiency for the photocatalytic reduction of Cr(VI) under visible-light illumination of other amino-functionalized MOFs (e.g., (Zr)UiO-66-NH_2_ and (Ti)MIL-125-NH_2_), commercial inorganic and organic photocatalysts (N-doped TiO_2_, g-C_3_N_4_), and even of the benchmark P25-TiO_2_ photocatalyst when irradiated under UV–visible light (Figure 11).

Fe-MOF hybrid photocatalysts have also been combined with carbon (rGO), g-C_3_N_4_, metal oxides (H_3_PMo_12_O_40_ (HPMo)), [9] WO_3_, TiO_2_, Bi_12_O_17_Cl_2_, and metal-halide and noble-metal nanoparticles (carbon quantum dots (CQDs)/Au, Ag, Pd, Pd, Pt, Au, and Ag—Table 2) to engineer advanced heterojunctions. The complexity of the heterojunctions has been steadily increased by (i) incorporating carbon/metal or metal halite/metal nanoantenna at the surface of the MOF particles, or by (ii) a direct physical mixture of MOFs and TiO_2_, metal/halides, and WO_3_ semiconductors or PANI polymeric conductors (Table 2). In parallel, (iii) the encapsulation of polyoxometalate species within the pore space of the MOF has been explored as well [248].

The construction of advanced heterojunctions based on carbon-based 2D materials has been explored with MIL-53 and MIL-101 materials. The doping degree of RGO or g-C_3_N_4_ is a key parameter to control in order to enhance the light-harvesting and photocatalytic efficiency of MIL-53, since surpassing a given threshold, the activity of these heterostructure materials starts to decline. MIL-101 has been modified with carboxylated g-C_3_N_4_ to improve the interfacial bonding of heterojunctions, reaching moderate chromium photoreduction efficiency in a third of the time compared to non-carboxylated systems.

In general, the heterostructure MOF-metal oxide photocatalysts exhibit better performance than carbon-based homologue materials. This is the case of MIL-53/WO_3_, MIL-100/HPMo, and MIL-100/WO_3_ materials, whose heterojunctions exhibit a better interplay to couple the photocatalytic degradation and adsorption functions (Table 2). It has been observed that the best catalytic performance of metal-oxide/MOF heterostructures is obtained at pH < pHpzc. The stability, the potential reusability, and the negligible effect of organic interferents for Fe-MOF/WO_3_ have been duly confirmed as well. For example, MIL-100/WO_3_ (80 wt.–120 wt.%) displayed outstanding stability and reusability during five successive cycling experiments on Cr(VI) photoreduction in synthetic water at pH 2 and enhanced Cr(VI) removal efficiency in the presence of low-weight organic molecules. It is important to mention that the same photocatalyst is partially inhibited with the presence of inorganic competing ions such as NO_3_^−^, Cl^−^, and SO_4_^2−^.

An alternative way to achieve Cr(VI) photoreduction in neutral-pH conditions has been reported by the modification of MIL-88B with TiO_2_ (i.e., 98.6% at pH 7 after 35 min of irradiation with visible light) (Table 2). In contrast, the modification of MIL-53 and MIL-100 with Bi_12_O_17_Cl_2_ improved the catalytic response under acidic-pH conditions for composites with a weight ratio (1:1) (Fe-MOF: Bi_12_O_17_Cl_2_). The weight ratio of MOF/metal oxides plays an important role in the modulation of the photocatalytic performance and tuning of the heterojunctions’ efficiencies, even if those are obtained from a physical mixture of the two components of the system (Table 2). The encapsulation of metal nanoparticles, or the direct crystallization of the metal nanoparticles at the surface of the MOF nanoparticles, offers an alternative approach to engineer advanced heterojunctions. This strategy also endows hetero-photocatalysts of hot-spots arising from the plasmonic functions of noble-metal nanosystems. That is, surface plasmons open the perspective to generating localized-heat points and active catalytic sites able to potentiate the photocatalysis in a synergic manner. For instance, the complete reduction of Cr(VI) under visible irradiation was achieved in short reaction times when combining MIL-100 with gold, palladium, or platinum metal nanoparticles (Table 2). Similarly, the Cr-MIL-101 photocatalytic efficiency for reducing Cr(VI) under visible-light illumination conditions has been improved by anchoring Pt, Pd, and Cu nanoparticles on its surface.

However, as is well known in metal-based catalysts applied for petrochemical applications, the agglomeration of the metal nanoparticles, or their size, induces a decay in the Cr(VI) photoreduction performance of these composite catalysts. Organic conductors as PANI polymers have also been mechanically integrated with MOF nanoparticles (i.e., MIL-100). The benefits arising from the combination of metal or organic-based electronic conductors into the framework of ordered porous materials have been duly demonstrated during the last decade. Nevertheless, it is important to note that we are far from fully understanding the underpinning mechanisms at the interphase/heterojunction between these two types of materials.

Overall, considering the state-of-the-art of Fe-MOFs for chromium photoreduction, the key parameters that potentiate their performance are: (i) acidic working conditions (i.e., pH < 4), (ii) the presence of hole-trapping agents (i.e., oxalic acid), (iii) amine functionalization of the frameworks, and (iv) the concentration and size of the metal oxide, metal, or carbon nanomaterials integrated within the MOFs to engineer their photo-response. Recent studies point out the benefits of integrating MOF materials in sand [239] and alumina (Table 2). Multivariate chemistry is and will be an active research area to boost the photoreduction efficiency of the MOFs over Cr(VI), as explained in the next section of this work (Table 2). An illustrative example is the bimetallic Fe/In MOF (MIL-68-NH_2_ (InαFe1-α)). The efficiency of these photocatalysts is highly dependent on the Fe(III)/In(III) content. For an α of 0.8, the reduction efficiency was lower even than the initial In-MOF, while an optimum balance between the cations, i.e., MIL-68-NH_2_ (In_0.4_Fe_0.6_), gave rise to 3.6 times faster photo-reduction than for MIL-68-NH_2_ (In) (Table 2).

### 3.3. Tetravalent-Metal-Based Metal-Organic Framework Photocatalysts

Tetravalent-based MOFs stand out in terms of chemical and hydrolytic stability in comparison with most of the trivalent- and divalent-metal-based homologues. Generally speaking, these background characteristics arise from the high connectivity of their archetypal inorganic building blocks: the poly-nuclear zirconium and titanium oxo-hydroxy clusters and chain-like subunit. Since the discovery of the archetypal UiO-66 and MIL-125 zirconium and titanium, their tailor function pre- and post-synthetic encoding has exponentially increased (Figure 12).

Surprisingly, the application of tetravalent MOFs for chromium photoreduction purposes has been limited to the benchmark UiO-66 and MIL-125 materials, and of their modifications. There is room for exploration in this specific research subject, since the structural and chemical versatility of tetravalent-based MOFs could open the perspective to understanding the effect of many key features of these materials (i.e., surface area, pore space, presence of defects, connectivity of the inorganic units, post-synthetic functionalization of the pore space, multivariate chemistry…) into their photocatalytic performance. For instance, for readers that could be interested in gaining a deeper understanding of the structural and chemical versatility of tetravalent-based MOFs, they may consult references [184,185,186,187,188,189].

Regarding the UiO-66 and MIL-125 materials employed for Cr(VI) photoreduction, they share a common topology arising from the similar connectivity of their inorganic and organic building units (Figure 12). Both compounds share the terephthalic-like linkers as their organic building blocks, and slightly differ in the characteristics of their inorganic building blocks. UiO-66 is built of hexanuclear [Zr_6_(μ_3_-O)_4_(μ_3_-OH)_4_]^12+^ clusters. The zirconium oxide nodes of UiO-66 can connect up to 12 carboxylate groups belonging to BDC linkers. Half of the eight oxygen atoms in the hydroxylated version of this SBU are bound to three zirconium atoms as individual atoms, and the remaining oxygen atoms are bound to three zirconium atoms in hydroxide form (Figure 12a).

UiO-66 crystallizes as a face-centered-cubic structure of F m−3m symmetry with a lattice parameter of 20.7 Å. The structure contains two types of cages: tetrahedron and octahedron pores of 7.5 Å and 12 Å, respectively. Ti-MIL-125 shares the same topology and connectivity of their inorganic and organic building units with UiO-66. Therefore, similar tetrahedral and octahedral cages and surface areas have been reported for this compound as well. The main difference lies on the inorganic nodes of the crystal structure. Titanium oxo-clusters in MIL-125 consist of a ring structure of eight-edge-shared Ti-octahedra capped by twelve carboxylate groups belonging to the BDC organic linkers. Overall, the connectivity of the Ti-oxo clusters (12-c) and the organic linkers (2-c) gives rise to an fcu topology with a slightly distorted tetragonal symmetry in comparison to the one of UiO-66. Readers may consult the recent review of Y. Bai et al. [249], L. Feng et al. [250], and Y. Chen et al. [251] for the crystal structure, defective chemistry, chemical and thermal stability, and potential applications of the UiO-66 family. Similarly, the investigation on MIL-125 has been intense since its discovery by Ferey’s lab that wrote specific reviews of titanium-based MOFs [252,253,254].

Taking into account the chemical stability of tetravalent metal-based MOFs, they have been started to be employed as photocatalysts for chromium photoreduction, as summarized in Table 3.

Regarding the application of UiO-66 for photocatalysis, most of the investigations have been focused on its chemical modification to prevent the recombination of electron–hole pairs and shift the photo-absorption edge from UV (3–5% of total sunlight) to the visible-light region. By applying a similar strategy reported for divalent and trivalent based-MOFs, Shen et al. (2013) [173] improved the Cr(VI) to Cr(III) photoreduction efficiency of UiO-66 by encoding amino groups into its framework (i.e., UiO-66-NH_2_). Due to the typical yellow color of the amino-terephthalic acid, and of the UiO-66-NH_2_ sample, the band gap energy was shifted to the visible-light region, unlocking the capacity of the material to drive the hexavalent chromium photoreduction under sunlight illumination. Extending this initial study, the same authors (Shen et al.) explored the light-harvesting and photocatalytic activity of UiO-66-NO_2_ and UiO-66-Br variants (Table 3).

The photoactivity of the UiO-66 frameworks was clearly related to their band gap energy, the amine variant being the one with the lower band gap energy and the most efficient one to photo-reduce chromium hexavalent species under visible-light illumination. Nevertheless, the correlation between the structure and the photoactivity of the UiO-66 frameworks was unclear. The metal substitution at the zirconium hexanuclear units of the UiO-66-NH_2_ framework has also been explored, both by post-synthetically doping the linker-defective positions of the structure with titanium ions [262]. It is interesting to note that for Ti/Zr-UiO-66, the improvement in the hexavalent chromium photoreduction does not arise from an energy band gap modification, since both Zr and Zr/Ti compounds exhibit similar light-harvesting characteristics, but from the improved photoconduction (i.e., reduced electron–hole recombination) of the Zr/Ti variant. Recently, UiO-66-(OH)_2_ has been revealed as the most efficient variant within the UiO-66 family to photo-reduce Cr(VI) to Cr(III). Its low band gap energy, together with its fast and efficient photoconductions, has given rise to a 100% conversion in less than 40 min under UV-Vis light illumination. It is important to note at this point that the hydroxyl variant of the UiO-66 framework shows an intermediate hydrolytic stability, so although the reusability tests indicate a negligible loss of activity, it would be key to assess to what extent the materials are releasing some of their inorganic or organic components to the media.

In comparison to UiO-66, the conduction band (CB) potential of the titanium-oxo cluster in MIL-125 is more positive than the ones reported for the zirconium hexanuclear clusters. This feature induces an efficient electron transfer from the photoexcited organic linker to the titanium-oxo cluster (Table 3). The main drawback of MIL-125 is its wide band gap, limiting its light absorption only to the UV region. Amine functionalization of the MIL-125 photocatalysts has been the first approach to shift its band gap to the visible-light region, arising from the electron donation of the N2p to the aromatic ring of the amino-terephthalate linker. In addition, the amino groups within MIL-125-NH_2_ act as a photosensitizer, improving the photocatalytic behavior in the reduction from Cr(VI) to Cr(III). It is well known that the electron transfer from the linkers to the Ti-oxo clusters generated Ti^3+^-Ti^4+^ pairs that play an additional role in transferring the electrons to the hexavalent chromium, or in generating radical oxidative or reductive species at the surface of the MIL-125 particles (Table 3).

Although the photocatalytic efficiency of MOFs has been widely proved, and the strategies to enhance their performance duly identified, until recently, their potential to work as dual sorbents/photocatalysts for the photoreduction and capture of Cr(VI) and Cr(III) has been widely overlocked. The first investigation in this regard was reported by P.G.-Saiz and coworkers for Ti and Zr benchmark UiO-66 and MIL-125 materials [46]. In order to elucidate the fate of Cr(III) ions during the photo-transformation of Cr(VI), the authors monitored both the Cr(VI) and Cr(III) concentration in the water solution during the Cr(VI) adsorption in dark conditions, and after triggering the photocatalysis through UV-Vis illumination.

Their findings demonstrate that even though MIL-125 was the best photocatalyst in terms of Cr(VI) reduction rate, the material was not able to fully retain the Cr(III) photo-transformed species. In contrast, the UiO-66-NH_2_ variant showed a full retention of the Cr(III) ions during photocatalysis, although the kinetics were slightly slower than those of MIL-125 (Figure 13).

It is interesting to highlight at this point a previous study of the same authors where they assessed the adsorption capacity of UiO-66 amine and hydroxyl variants for Cr(VI) and Cr(III) adsorption [135]. In this work, they reported an experimental approach to determine the Cr(VI), Cr(V) and Cr(III) speciation within the MOFs after adsorption. This approach can be easily adapted in the future to study the chromium speciation evolution during photocatalysis as well (Figure 14).

The engineering of advanced heterojunctions has been also applied for zirconium and titanium MOFs. Shen et al. [28] induced an electrostatic self-assembly of UiO-66-NH_2_ and GO, and posteriorly, reduced GO to rGO via hydrothermal treatment. The heterostructured UiO-66-NH_2_/rGO improves the visible-light absorption and the efficiency to separate the photo-generated electron–hole pairs, due to the electron conductivity of graphene functionalities. Overall, UiO-66-NH_2_/rGO exhibits significantly improved photocatalytic activity under visible-light illumination in comparison to UiO-66-NH_2_.

Similarly, UiO-66/C_3_N_4_ composites showed a limited recombination of photo-induced charge carriers due to the enhanced mobility of photogenerated electrons induced by g-C_3_N_4_ sheets (Table 3).

In comparison to its zirconium counterpart, the heterostructured photocatalysts constructed from MIL-125 materials have been mainly based on metal oxide or metal sulphide nanoparticles. The formation of a heterojunction system consisting of narrow-gap semiconductors, such as MoS_2_, Ag_2_S, CdS, and CuS on MIL-125, shifts the absorption of the heterostructured materials to the visible region (Table 3). In this case, metal sulphide nanoparticles also have a similar sensitizer effect to that NH_2_ encoding into the framework [263,264]. Additionally, E. Dhivya et al. (Table 3) reported the synthesis of a heterostructure system involving two Ti-based MOFs (NH_2_-MIL-125 and NTU-9 (Ti)) for increasing the charge separation. In this case, the 1,4-dioxido-2,5-benzenedicarboxylate organic linkers of NTU-9 expand even more the light-harvesting capacity of the system to the visible-light region (i.e., 2.54 eV and 1.29 eV band gaps). The combined system exhibits the highest photocatalytic performance in Cr(VI) reduction due to the efficient transfer of the photogenerated electrons on the charge band of NTU-9 to the empty valence band of MIL-125-NH_2_.

## 4. Multivariate Metal-Organic Frameworks for Chromium Photoreduction

Multivariate reticular chemistry offers an opportunity to tailor and balance the light harvesting, photoconduction capacity, and oxygen radicals generation to achieve a fast and efficient chromium photoreduction. The variance of the different functional groups encoded within the ordered pore space of MOFs opens the avenue to obtain synergistic effects. For instance, multivariate MOFs (MTV-MOFs) possess more than two functionalities randomly distributed within the framework that work together in a cooperative or coupled fashion, outperforming—as an ensemble—their homogenous and periodic counterparts [264,265,266].

MTV-MOFs must not be mixed up with multicomponent MOFs, where the multiple linkers are topologically different from one another in terms of length and connectivity, and thus, can be distinguished individually in a crystalline lattice. Indeed, the fundamental criteria of MTV-MOFs are specific functionalities occupying a similar location in the framework and a changeable percentage of each functionality. This way, the introduction of varied functional groups can be achieved without altering the underlying backbone of their structure, obtaining a “heterogeneity within the order”.

MTV-MOFs are classified as mixed-ligand (ML) and mixed-metal (MM) MOFs. Just as an illustrative example of the versatility of MTV materials, in 2010, O. M. Yaghi et al. reported the first ML-MTV-MOF, incorporating a terephthalate linker and its eight derivatives within one pure phase of a MOF-5 compound [267]. Since then, the application of MTV-MOFs has been expanded to many research areas, including Cr(VI) photoreduction. It is important to mention at this point that X. S. Wang et al. [268]. Have recently reviewed the application of MTV-MOF materials for chromium photoreduction purposes. Below, we have tried to highlight these works focused on MTV-MOF reticular materials that have been published after the seminar compilation developed by the abovementioned authors.

Recently, Valverde et al. [136] designed a multivariate UiO-66 to develop dual sorbent photocatalysis for the removal of Cr(VI) in wastewater. Many studies have explored how replacing the original terephthalate (TPA) linker of the UiO-66 framework for some of its derivatives can endow the material with chemical (i.e., dihydroxyterephtalate (TPA-(OH)_2_)) and photocatalytic (i.e., aminoterephtalate (TPA-NH_2_)) capacity to reduce Cr(VI) to Cr(III), as with the chemical affinity to adsorb both Cr(VI) (i.e., TPA-NH_2_) and Cr(III) (i.e., TPA-(OH)_2_). However, both UiO-66-NH_2_ and UiO-66-(OH)_2_ lack the chemical robustness to work under highly acidic or caustic conditions that only the nitro-functionalized UiO-66-NO_2_ can tolerate (Figure 15). Multivariate reticular chemistry offers an opportunity to tailor and balance all the targeted characteristics to achieve a fast and efficient Cr(VI) to Cr(III) photoreduction via the synergistic combination of different functional groups. In this study, Valverde et al. employed multivariate functionalization strategy to tune multiple chemical characteristics of the UiO-66 structure, such as the light harvesting, the adsorption capacity over Cr(VI) and Cr(III) species, photoconduction efficiency, and Cr(VI) to Cr(III) chemical reduction and photoreduction properties (Figure 15).

In parallel, they explored how the compositional variance in MTV-MOFs affects their hydrolytic stability in comparison to the one of their parent single-functionalized frameworks. From the overall performance to photo-reduce and capture chromium ions, the balanced multivariate functionalization of the UiO-66-NH_2_/-(OH)_2_/-NO_2_ framework resulted in a dual sorbent/photocatalyst with: (i) efficient chemical/photo-reduction of Cr(VI) to Cr(III) and (ii) retention through adsorption of the resulting Cr(III) ions.

Regarding the mechanisms for the chemical and photocatalytic transformation and immobilization of chromium in the UiO-66 MTV-MOFs, it is important to note that the modification of the chromium oxidation state is linked to variation in its coordination environment. Overall, Cr(VI), stabilized as CrO_4_^2−^ chromate anions, gains three electrons and incorporates two hydroxyl or water molecules within its coordination environment during its reduction to Cr(III). During this process, the highly reactive and transient intermediate Cr(V) species are formed as well, as A. Valverde et al. [136] proved through EPR. The authors stated that the first step of the immobilization and transformation of Cr(VI) to Cr(III) into the UiO-66 frameworks is the adsorption of chromate anions (Figure 16a,b). Two possible mechanisms explain the chromate adsorption capacity of the UiO-66 frameworks, their covalent immobilization to the linker-defective positions located at the zirconium hexanuclear clusters, or their electrostatic interaction with hydroxyl, but especially, with amine-protonated groups. Reached at this point, two possible paths for the Cr(VI) to Cr(III) reduction are possible: (i) photocatalysis and (ii) chemical reduction. The chemical encoding of the UiO-66 frameworks determines the efficiency and combination of the separated paths.

During photocatalysis, the light-harvesting capacity of the UiO-66 frameworks (i.e., band gap) promotes the electron and hole separation. Nevertheless, the conduction and transfer of electron/hole pairs are governed by the photoconduction efficiency of the materials (Figure 16(c.1)). At this point, the chemical variance of MTV materials makes the difference. In fact, they realized that the incorporation of an electron donor and withdrawing groups within the same framework improves the photoconduction and fastens the Cr(VI) to Cr(III) photo-transformation. The chromate anions stabilized within the porous scaffold are steadily reduced to Cr^V^ and Cr(III) while they are immobilized into their adsorption position (Figure 16(c.2)). As the photocatalytic transformation of Cr(VI) to Cr(III) evolves, the photoreduction process is repeated, leading to the clustering of Cr(III) ions within the framework (Figure 16(c.3)). It is important to note that if the single or MTV-UiO-66 lacks hydroxyl groups, transient Cr(V) species will be stabilized within the material after operation. In contrast, when hydroxyl functionalities are encoded in UiO-66, the Cr(VI) adsorption process (Figure 16(b,d.1)) is coupled to its chemical reduction to Cr(III) via electron-rich quinone groups coming from hydroxyl functionalities (Figure 16(d.2)). Finally, they stated that as the chemical reduction, or its combination with photocatalysis, evolves, chromium ions are stabilized as clustered Cr(III) ions with the frameworks (Figure 16(d.3)). Regardless of whether photocatalysis, chemical reduction or their combination is the process that triggers the Cr(VI) transformation to Cr(III), the presence of hydroxyl ions is key to destabilizing Cr(V) transient species and transforming them into Cr(III) ions. This does not preclude the presence of Cr(VI) and Cr(III)ions within hydroxyl-functionalized frameworks, but the absence of highly reactive and toxic Cr(V).

MM-MTV-MOFs are complex to synthetize. For these materials, different metals are mixed in the inorganic secondary building units (SBU). However, this often results in the synthesis of mixed MOF phases, rather than a single MM-MTV-MOF. This issue can be overcome by choosing metals with similar valences and making sure that they can form the same SBU. Another strategy to overcome this issue is the transmetalation of the material. With this post-synthetic modification, heterometallic MM-MOFs can be obtained, which cannot be achieved through normal synthesis because of the different reactivity of the metal ions [269,270]. MM-MTV-MOFs are highly promising as photocatalysts, since they offer two metal reaction centers in apparent proximity to the distinct photocatalytic performance. Even if they have not been studied yet for Cr(VI) photoreduction, they have been shown to enhance the photocatalytic activity to reduce CO_2_. Indeed, the presence of a second metal as an electron mediator in these materials promotes EHP transference from the excited state of the linker to metal ion clusters [271]. Sun et al. [272] introduced Ti via post-synthetic metal exchange in (Zr) UiO-66-NH_2_, and their material showed enhanced photocatalytic performance for both CO_2_ reduction and hydrogen evolution under visible light.

MM-MOFs and ML-MOFs can also be merged in a single material. For instance, Navarro Amador et al. [273] obtained a material based on (Zr) UiO-67 with mixed ligands and mixed metals. The synthesis of the material was made through the combination of two synthetic pathways: first, the solvothermal synthesis with two different linkers (the original linker biphenyl-4,4′-dicarboxylic acid, and a similar one modified with Ru to be used as light antenna). Afterwards, Ti was included via post-synthetic metal exchange on the coordination node of UiO-67, and a material able to remove organic pollutants from an aqueous solution and to catalyze the degradation of the pollutant under visible-light irradiation was obtained. Concretely, the material showed to be active towards the degradation of methylene blue with a good improvement due to the modifications on the structure even when the exchange was not complete, proving the interaction between the light antenna and the catalytic center, since the materials with just one of the mentioned modifications did not show a big improvement in the catalytic activity. These studies show the promising materials that can be obtained via the multivariate strategy, and that could be extended to Cr(VI) photoreduction soon.

## 5. Future Perspectives of MOFs for Chromium Photoreduction

We wish to place the research of MOFs for Cr(VI) photoreduction into perspective with the forefront advances in (1) reticular chemistry, (2) materials for catalysis and photocatalysis in environmental remediation [274,275,276], and (3) their hybridization/integration into heterostructured systems with improved photochemical properties [277,278,279,280]. In parallel to the fundamental perspective, it is of paramount importance to keep the application perspective of the commercialization of MOFs for photocatalysis in real scenarios firmly in mind (Figure 17) [281,282].

First, we wish to highlight that there is still plenty of room for innovations for MOF material design in order to optimize chromium detoxification. From the vast MOF compounds available to date, only few with similar inorganic and organic building blocks have been tested for Cr(VI) detoxification purposes [283]. The scope of MOFs built up or functionalized with chromophore-like organic linkers [284,285] or metal linkers [286] tested for chromium photoreduction are highly scarce in comparison to the current scope of possibilities. Many of these have been specifically designed for reductive catalysis or photocatalysis purposes far from chromium reduction, such as carbon dioxide transformation [287,288] or water-splitting purposes [289]. Defect engineering or post-synthesis are mostly unexplored strategies so far for MOF material design for chromium detoxification [290,291,292]. From the varied strategies proposed to install metal-catalytic and photocatalytic sites into MOFs after their synthesis (i.e., chemical vapor deposition [293], metal adsorption [294], and transmetalation), just the most conventional ones have been explored so far. A similar scenario is found when multivariate or multicomponent reticular materials are considered [295,296]. Mixed-metals/linker or multicomponent MOFs could make the difference in terms of light harvesting, exciton generation, and hole and electron separation and transport. For instance, the compositional and structural variances within MTV-MOFs have already led to synergistic functions outperforming the efficiency of singly functionalized MOF variants for gas adsorption, [297] drug release, and even for photocatalysis purposes, including a few works which have been published exploring their potentials for chromium photoreduction.

Similarly, the hybridization/integration of MOFs into heterostructured systems applied for Cr(VI) photoreduction is still in its infancy, since from the multiple options (Figure 4), only three have been explored deeply (i.e., surface decoration and core–shell structures). The complexity of the heterostructured materials based on MOFs designed for photocatalytic applications in general far exceeds the ones developed for chromium photoreduction. In addition, the integration (i.e., structure and interphase between the different materials conforming to the heterostructure) is key to modulate the photochemical response of the hybrid heterojunction. For instance, physical or chemical in situ growth, or an interphase specifically designed to facilitate a smooth transition between the materials forming the nanostructure, is key to modulating the overall photochemical properties (among other ones) of the system. This point has been carefully studied when integrating the MOFs as surface thin films for gas membrane separation, or is in place to integrate the MOF properly in electronic devices or signal-transducer materials [298,299,300]. For instance, self-assembly [301,302], printing [303], deposition and patterning of MOFs [304] can play a key role to endow them with photonic properties to adsorb specific wavelengths [305,306,307,308,309,310,311]. In addition, the heterojunctions at this thin-film scale have been revealed to be beneficial for photocatalytic purposes as well.

As mentioned before, these advances in fundamental MOF design and integration levels need to be contextualized considering their final environmental applicability. In contrast to catalysts for petrochemical or fine-chemical purposes, the following has to be considered for a material used in environmental remediation: (1) the chemical stability, (2) eco-toxicology, (3) reusability, and (4) production costs and large-scale synthesis feasibility.

Regarding the chemical stability, there is still room for improvement to evaluate these parameters. Current research works fail to fully evaluate the hydrolytic and chemical stability of MOFs, since most of them do not quantify or estimate the leakage of the MOF components (metals, organic linker, or functional group molecules incorporated to the framework) during operation. For instance, the long-term hydrolytic stability of most of the MOFs explored for chromium photoreduction is questionable, especially in the case of divalent-metal-based carboxylate frameworks [312,313,314].

From an eco-toxicological perspective, this issue could be partially solved if non-ecotoxic components were used to build up the MOF photocatalysts [315]. For example, iron, titanium, and zirconium (to a lesser extent) and fumaric acid, aspartic acid, or tartaric acid could be very interesting building blocks to find a MOF solution with negligible environmental effects if any leaching occurs during operation. Iron-based MOFs are especially appealing in this regard, since besides being built up from an environmentally friendly, cheap, and abundant metal, high-yield water green synthesis routes have been developed during the last decade to obtain them at large scales. If assembled from environmentally friendly linkers such as fumaric acid (i.e., MIL-88, MIL-101…), the ecotoxicological impact of Fe-MOFs could be greatly reduced even if the materials are not as hydrolytically stable as their chromium or zirconium homologues. Nevertheless, it is important to note that the ecotoxicology of the MOF materials themselves needs to be evaluated as well to discard any negative environmental impact if MOF particles are leached.

It is difficult to evaluate and balance by conventional means the structural, porous, stability, performance and ecotoxicity features that are needed to achieve a MOF system to be applied for chromium photoreduction in real application conditions. Here is where machine learning could play a disruptive role during the coming years in order to identify, screen, classify, and correlate MOFs’ potentials on the basis of geometric, chemical, topological, energetic, and performance-based descriptors [316,317]. In fact, machine learning is already having a deep impact on unraveling synthesis paths and engineering strategies of MOFs for gas adsorption and separation purposes [318,319,320,321]. As far as the investigations of MOFs for photo-oxidative and photoreductive processes expand, it is more likely that machine learning could be applied to unravel the underpinning chemical–physical features that make the MOFs feasible for this application.

The integration, recovery, and reactivation of MOF catalysts within applicable photocatalytic systems are the main objectives. The immobilization of the MOF powdered materials on sand or alumina could help to scale up the MOF application for water remediation, especially in terms of the hydraulic conductivity of the final system. An economical and easily accessible strategy could be the integration or immobilization of MOF sorbent/photocatalysts into metal oxide, ceramic, or polymeric membranes and filters. In addition to the possible synergic effects to combine two semiconductor materials in a final device form, once supported/integrated on different substrates, the time and energy consumption for the recovery and reactivation of the MOF would be significantly reduced. These MOF/integration strategies need to consider as well the MOF shaping and filming technologies that have been developed during the last decade [322]. The hybridization of MOFs with polymers has been a natural research step that has been applied for the heavy-metal detoxification of water, but that has been rarely tested for chromium adsorption and or photoreduction purposes [323,324,325]. Similarly, the growth of MOFs as surface porous continuous layers onto varied supports has given rise to a portfolio of thin-film porous materials with optical and transport properties far from those of the bulk materials. Still, there is plenty of room to investigate in this regard when applying the MOFs for chromium photoreduction purposes. In parallel, shaping the MOFs as pellets and foams of 3D-printed objects with added meso-macroporosity features could close MOFs to real technologies such as water detoxification without losing the intrinsic functionality of the parent material [326,327,328].

Last but not least, up to now, most of the MOF materials are usually produced at the laboratory scale, and their validations in the photoreduction of chromium have been carried out in synthetic solutions. Thus, the feasibility of massively producing these materials using low-cost metal centers on a larger scale without affecting their properties and the cost-effectiveness is necessary. An advantage of MOFs in comparison to alternative photocatalysts or adsorbents is that their feasibility to capture and degrade organic pollutants—even persistent organic pollutants or chemicals included within the priority watch list of the European commission such as drugs, hormones or pesticides—has been corroborated at the lab scale [329,330,331]. Thus, the development of advanced, cheap and environmentally friendly MOF photocatalysts does not only open the perspective to face hexavalent chromium pollution, but alternative source of organic and inorganic hazardous chemicals that are usually found in wastewater streams.

In this sense, the evaluation of MOF technologies to face different types of pollution sources in complex water matrices could make the difference compared to traditional sorbent or photocatalyst materials. Overall, the drawback of the cost of producing MOF materials could be mitigated if their feasibility to capture or degrade pollutants that traditional technologies fail to remove is proved both at the lab scale and at close-to-real conditions. For instance, even if the costs arising from MOF production could prevent their application in large-scale wastewater installations, they could be a technically and economically feasible solution to develop portable water remediation systems applicable in isolated areas.

All in all, reticular chemistry offers a plethora of opportunities to function-tailor MOF materials for chromium water remediations, but also for photocatalytic and adsorption purposes in general. Going from the lab bench to real-world applications, the strategies designed at lab-scale to engineer and improve the performance of MOFs need to be balanced with their cost of production and environmental impact in terms of their ecotoxicity or the ecotoxicity of their components. The point that could make the difference with their commercial counterparts could be the possibility to face the chromium photoreduction concurrently with specific sources of pollution that traditional technologies fail to remediate.

## Figures and Tables

**Figure 1 nanomaterials-12-04263-f001:**
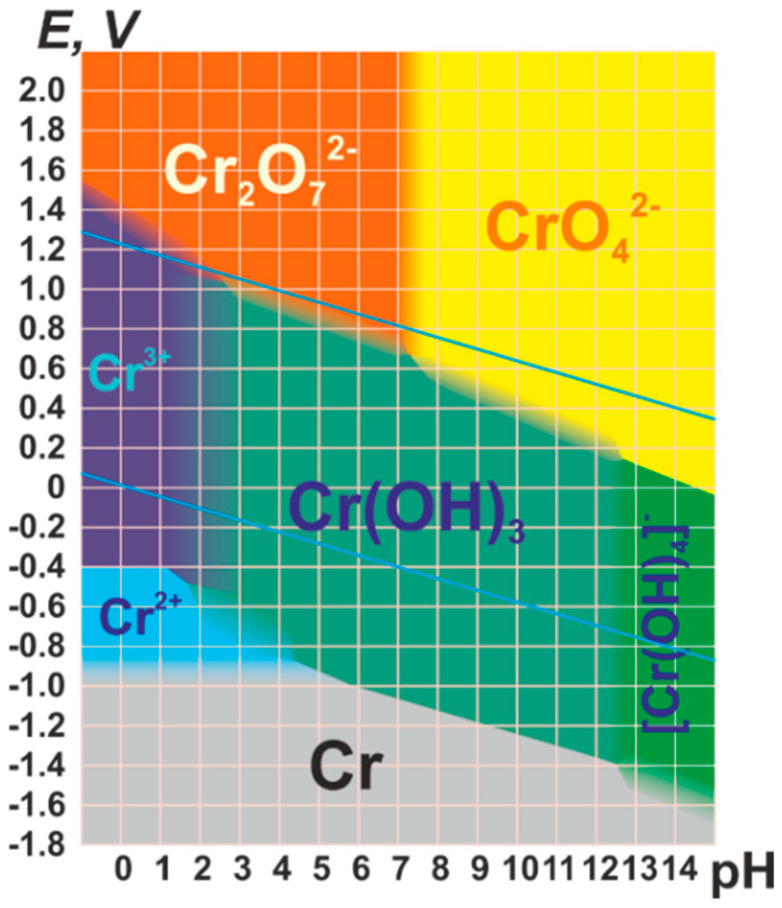
Pourbaix diagram of Cr. Reproduced from Denis Zihilin. Copyright CC BY-SA 3.0 License: https://creativecommons.org/licenses/by-sa/3.0/legalcode. “URL (accessed on 1 November 2022)”.

**Figure 2 nanomaterials-12-04263-f002:**
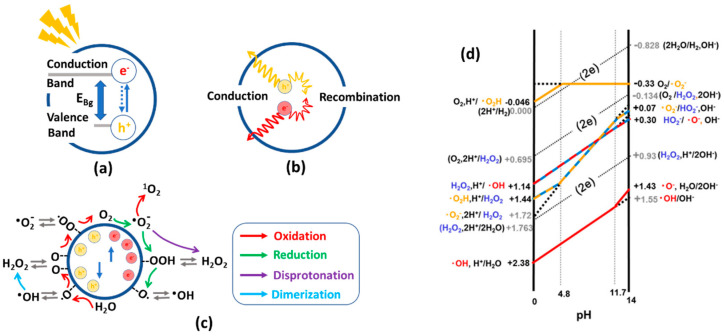
Scheme of photocatalytic process on semiconductors. (**a**) Illustration of the optical band gap (EBg) of a semiconductor and the separation of electrons and holes during illumination. (**b**) Transport and recombination of electron and hole pairs in a photocatalyst. (**c**) Oxygen reactive species generated due to the oxidation, reduction, deprotonation and dimerization reactions at the surface of the photocatalysts. (**d**) pH dependence of one-electron redox of H_2_O, H_2_O_2_, and O_2_. Dotted line shows two-electron (2e^−^) process. Reproduced with permission of the American Chemical society from reference [8].

**Figure 3 nanomaterials-12-04263-f003:**
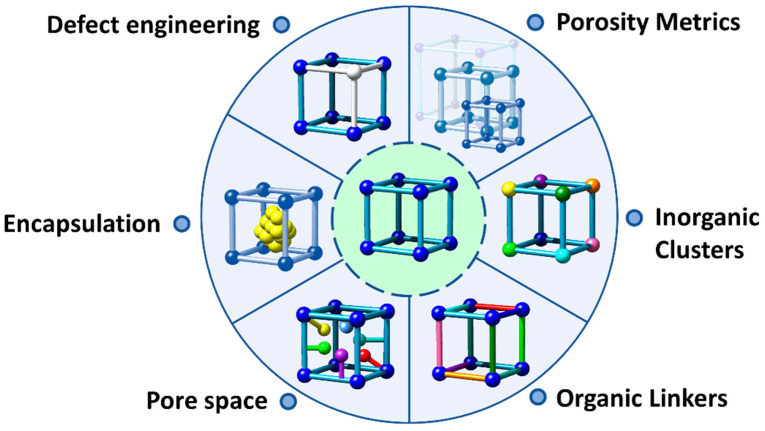
Main functionalization strategies applied to modify the framework and pore space characteristics of MOF materials for water photocatalytic remediation purposes [24].

**Figure 4 nanomaterials-12-04263-f004:**
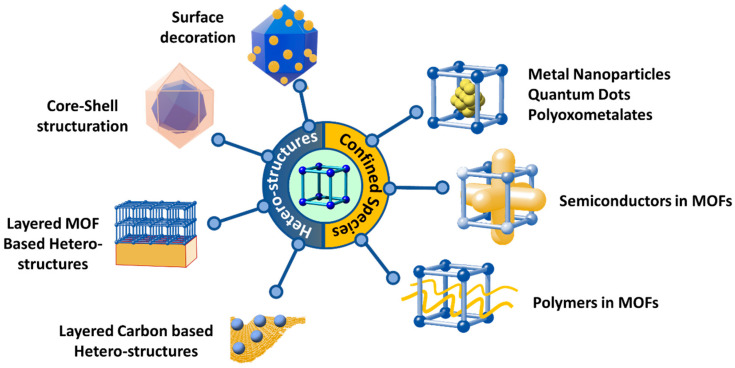
Main strategies investigated to engineer advanced heterojunctions based on MOF-heterostructured materials.

**Figure 5 nanomaterials-12-04263-f005:**
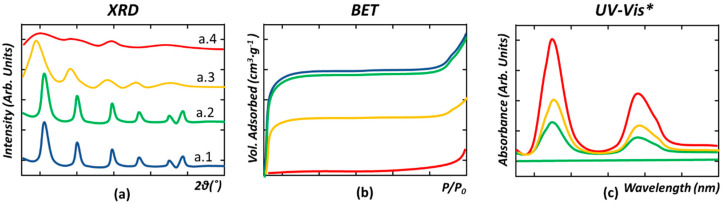
Illustration of the characterization techniques applied to determine the chemical stability of MOFs. (**a**) X-ray diffraction patterns and (**b**) N_2_ adsorption isotherms of chemically robust-to-unstable MOF materials. (**c**) UV-Vis spectra of the water solution after the leakage of organic linkers of robust-to-unstable MOFs. * The UV-Vis spectra is done at the solution where the MOF material has been in contact with, and not to the MOF material itself. The following color code has been used to depict MOFs with different degrees of chemical stability: Blue—very robust, green—robust, yellow—intermediate, red—unstable.

**Figure 6 nanomaterials-12-04263-f006:**
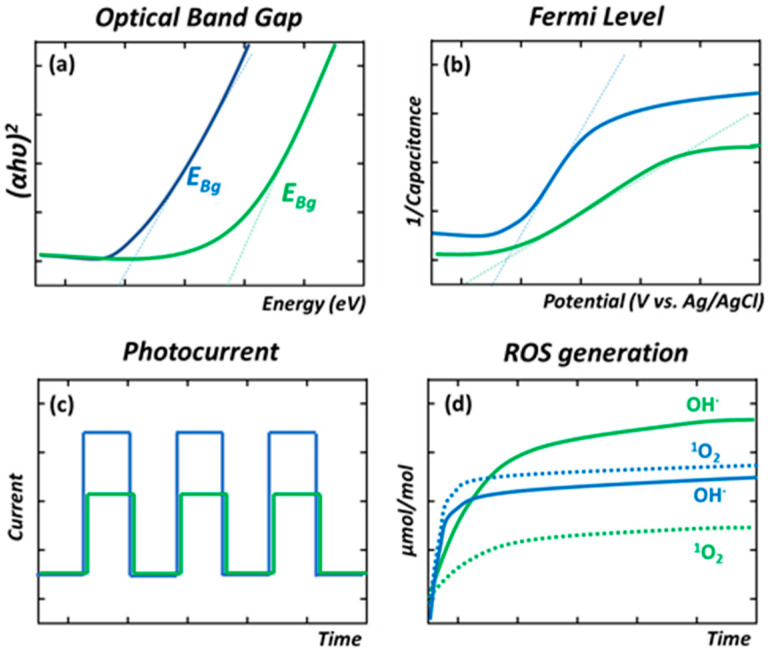
(**a**) Optical band gap energy calculation based on the Tauc plot fitting. (**b**) Determination of the Fermi energy level via the fitting of the Mott–Schottky plot. (**c**) Photocurrent response of the photocatalysts when illuminated. (**d**) Evolution of the cumulative quantity of reactive oxygen species during illumination of the photocatalysts. All the characterization protocols are illustrated with the hypothetical response of two model MOFs shown by blue- and green-colored curves. Dashed lines in (**a**,**b**) plots represents the fitting of the experimental data to determine the optical band gap and the Fermi levels of the photocatalysts. The solid and dashed lines in figure (**d**) represents the generation of hydroxyl and superoxide radicals when the material is illuminated.

**Figure 7 nanomaterials-12-04263-f007:**
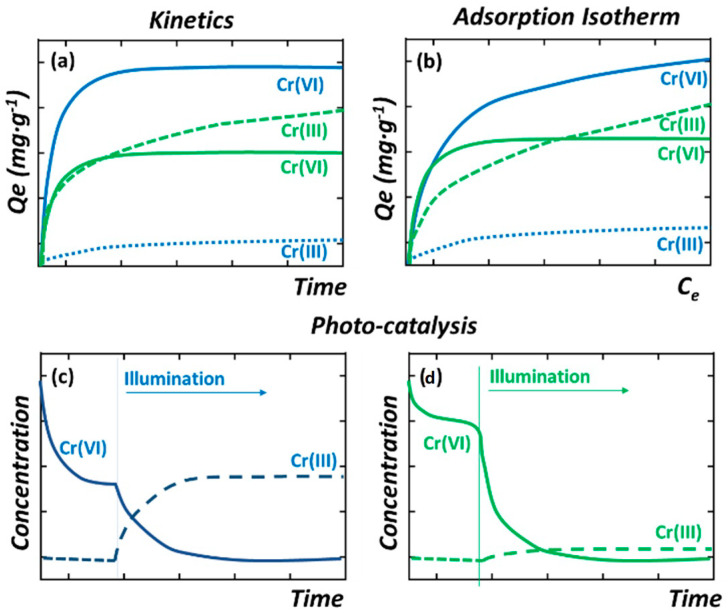
(**a**) Optical band gap energy calculation based on the Tauc plot fitting. (**b**) Determination of the Fermi energy level via the fitting of the Mott–Schottky plot. (**c**) Photocurrent response of the photocatalysts when illuminated. (**d**) Evolution of the cumulative quantity of reactive oxygen species during illumination of the photocatalysts. Blue-colored curves illustrate the response of a model MOF with a high adsorption capacity over Cr(VI) and a negligible adsorption affinity over Cr(III). Opposite, green-colored curves illustrate the performance of a MOF able to capture both Cr(VI) and Cr(III) species.

**Figure 8 nanomaterials-12-04263-f008:**
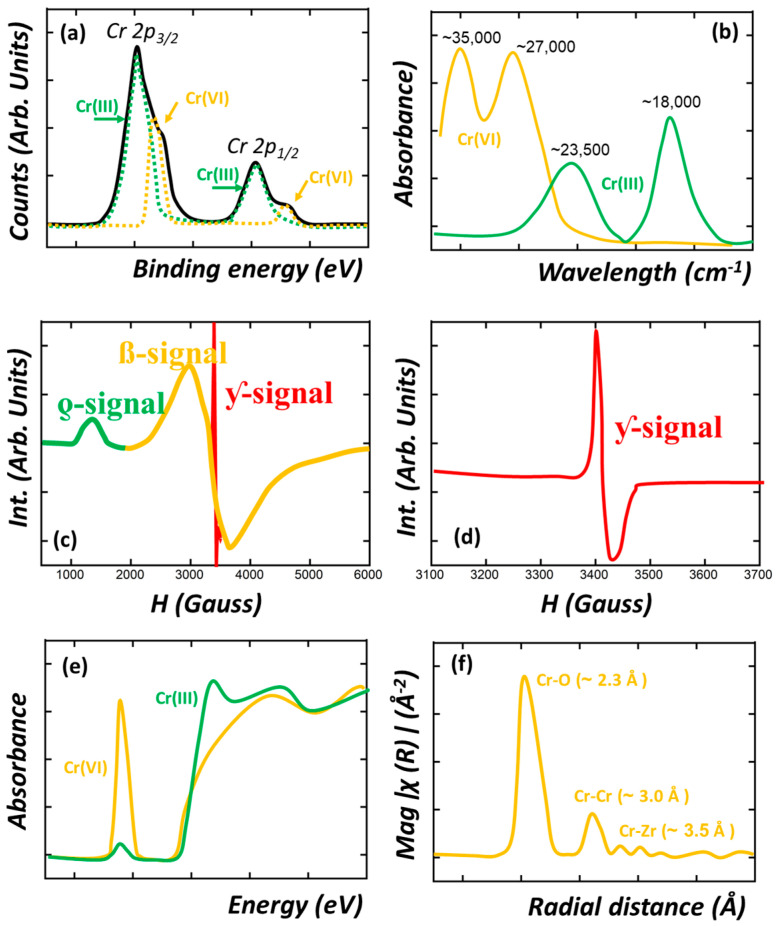
Illustration of the (**a**) XPS and (**b**) UV-Vis spectra of Cr(VI) and Cr(III) species stabilized within MOFs. (**c**,**d**) EPR spectral fingerprints arising from Cr(III) (**c**) and Cr(V) (**d**) species stabilized within MOFs after adsorption and photocatalysis. (**e**) X-ray adsorption near-edge structure of Cr(VI) and Cr(III) species, and the illustration of (**f**) the radial distances’ distribution obtained from the treatment of X-ray absorption data. Green-, orange-, and red-colored lines have been used to illustrate the signature of Cr(III), Cr(VI), and Cr(V) ions in the different plots. Dashed lines in the figure (**a**) stands out for the fitting of the XPS data to the contributions of Cr(VI) and Cr(III) ions.

**Figure 10 nanomaterials-12-04263-f010:**
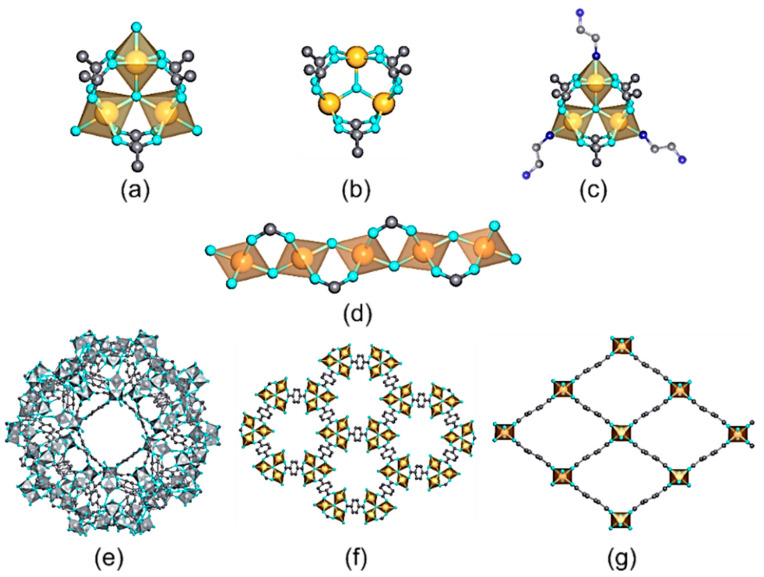
Inorganic structural units found in trivalent MOFs used for metal ion recovery in aqueous media. (**a**) M_3_(µ_3_-O)(R─CO_2_)_6_AlS_2_ (A = Cl, OH, F) trimers; S = Solvent (**b**), M_3_(µ_3_-O)(R–CO_2_)_6_A_1_ trimer after solvent removal (**c**), M_3_(µ3-O)(R–CO_2_)_6_Al(en)_2_; trimers after their decoration with en (ethylenediamine) molecules, (**d**) [M(µ_2_-A)(R–CO_2_)_2_]_n_ chains. Crystal structures of (**e**) MIL-100, (**f**) MIL-88, and (**g**) MIL-53 materials.

**Figure 11 nanomaterials-12-04263-f011:**
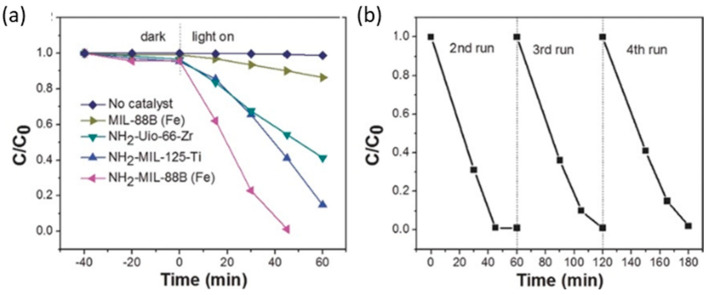
(**a**) Photocatalytic reduction of Cr(VI) over various Fe(III), Zr(IV), and Ti(IV)-MOF-based photocatalysts; (**b**) Photoreduction performance of NH_2_–MIL-88B (Fe) for four consecutive reactivation and utilization cycles. Reproduced with permission from reference [161].

**Figure 12 nanomaterials-12-04263-f012:**
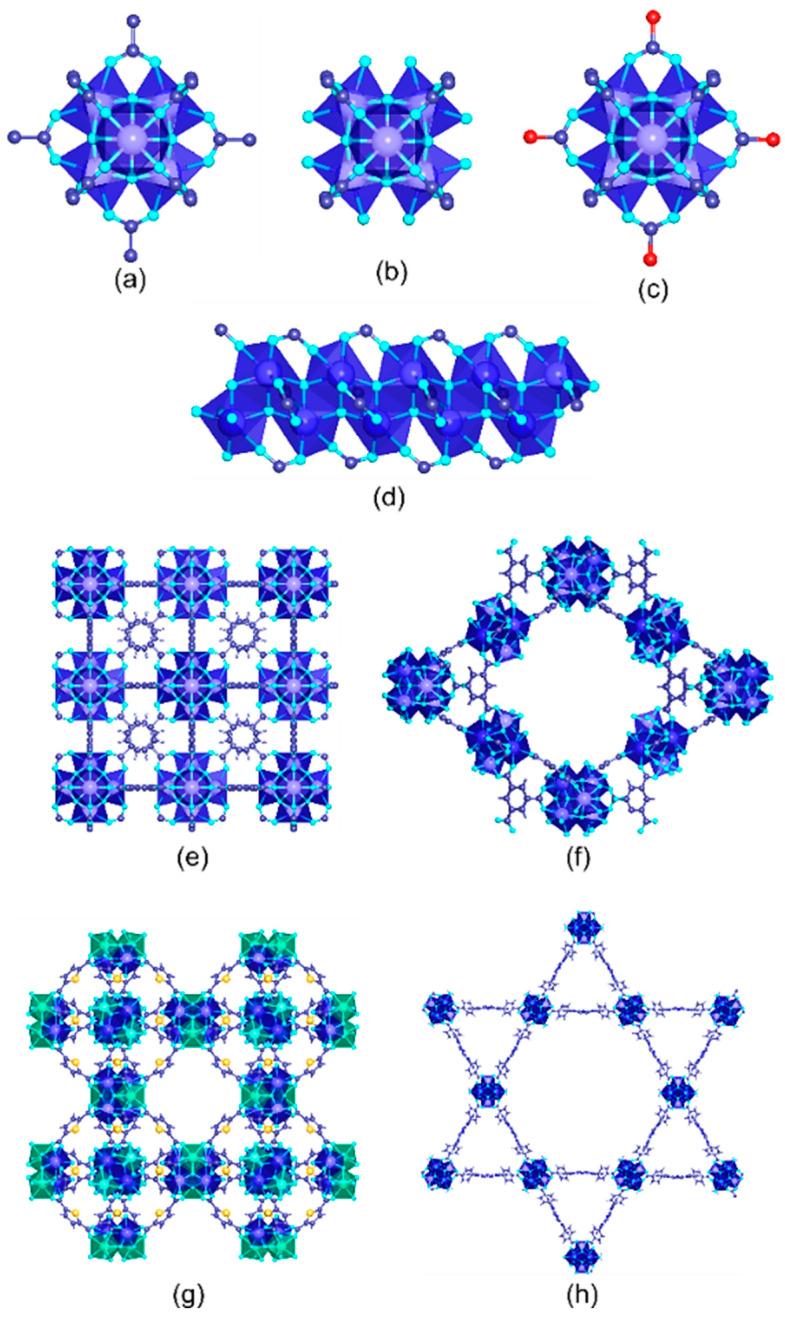
Inorganic structural units found in tetravalent zirconium MOFs used for metal ion recovery in aqueous media. (**a**) Zr_6_O_4_(OH)_4_(CO_2_)_12_, (**b**) defective Zr_6_O_4_(OH)_4_(CO_2_)_12_, and (**c**) functionalized Zr_6_O_4_(OH)_4_(R_1_-CO_2_)_12-X_(R_2_-CO_2_)_X_ hexanuclear clusters. (**d**) Zr one-dimensional units found in MIL-140 structure. Crystal structures of the zirconium MOFs used for metal ion adsorption, (**e**) UiO-66, (**f**) MOF-808, (**g**) DUT-67, (**h**) MOF-545.

**Figure 13 nanomaterials-12-04263-f013:**
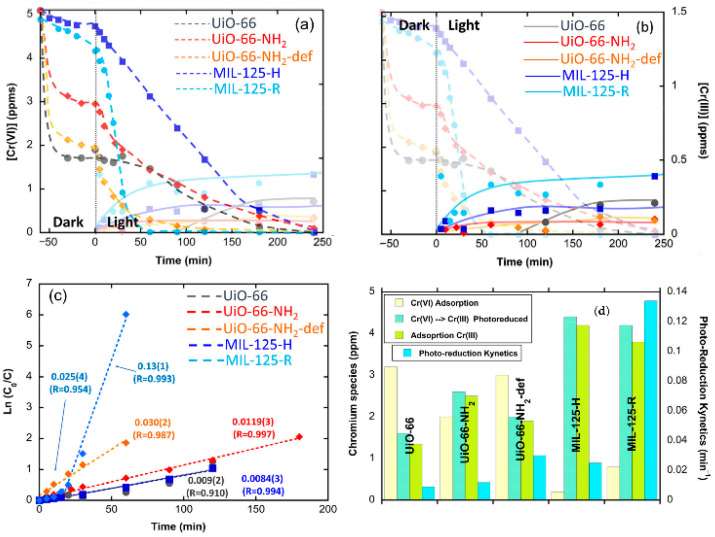
(**a**,**b**) Adsorption and photocatalytic reduction of Cr(VI) in the different MOF samples under UVA light: (**a**) Detail of the Cr(VI) and (**b**) Cr(III) concentration evolutions. (**c**) Fitting of the photoreduction kinetics. (**d**) Summary of Cr(VI) adsorbed at the MOF at dark conditions, the total amount of Cr(VI) photoreduced to Cr(III), the amount of photoreduced Cr(III) adsorbed at the MOF, and the photoreduction rate of the studied materials. Reproduced with permission from reference [64].

**Figure 14 nanomaterials-12-04263-f014:**
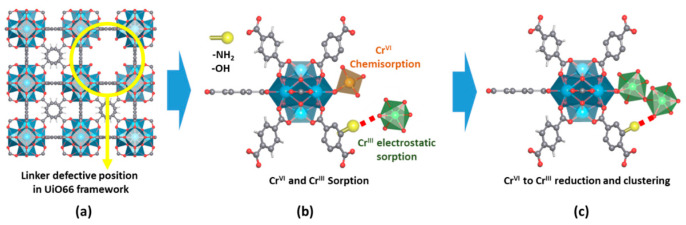
(**a**) Local structure of a linker defect position in UiO-66. (**b**) Detail of the possible Cr(VI) and Cr(III) adsorption positions at an under-coordinated defect position and organic linkers of zirconium hexanuclear clusters. (**c**) Cr(VI) to Cr(III) reduction induced by the presence of electron-donor groups at the organic linkers or photoactivity of the material. Reproduced with permission from reference [65].

**Figure 15 nanomaterials-12-04263-f015:**
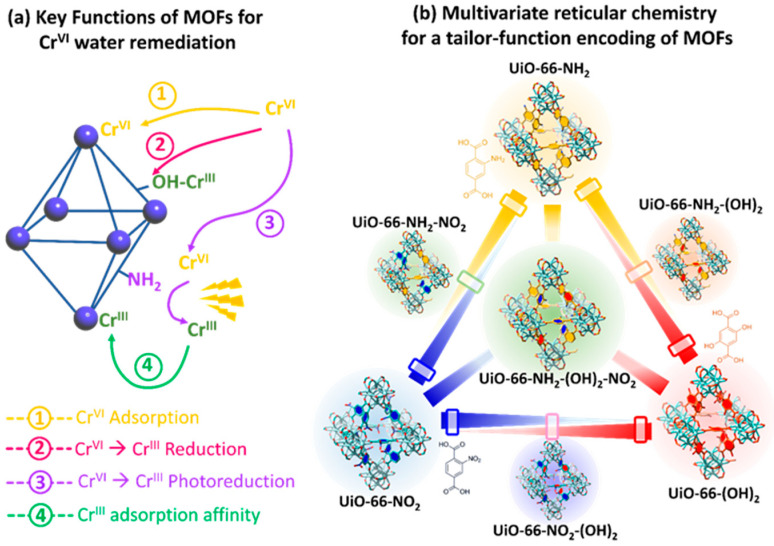
Illustration of the potentials of MTV-MOFs designed for improved chromium photoreduction and adsorption.

**Figure 16 nanomaterials-12-04263-f016:**
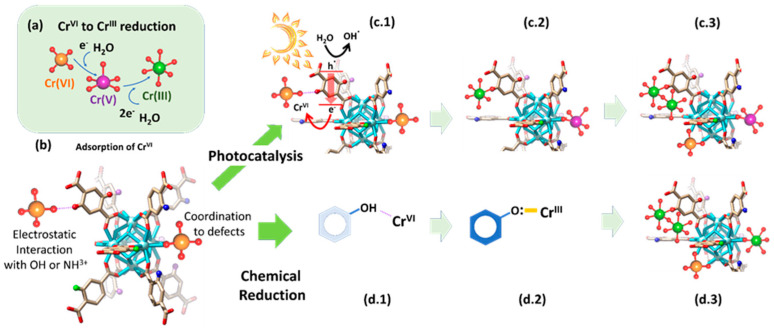
Cr(VI) to Cr(III) chemical and photoreduction mechanisms in MTV-UiO-66 dual sorbent/photocatalysts. (**a**) Overall Cr(VI) to Cr(III) reduction including Cr(V) intermediate species. (**b**) Adsorption mechanisms for Cr(VI) oxyanions within the MTV-UiO-66 frameworks. (**c.1**) Light-triggered generation of electron and holes and the concurrent electron transfer from the UiO-66-R framework to Cr(VI) oxyanions. (**c.2**,**c.3**) Evolution of the chromium species into the porous frameworks during photocatalysis. (**d.1**) Electrostatic interaction between hydroxyl groups and Cr(VI) ions. (**d.2**) Chemical reduction of Cr(VI) to Cr(III) and their stabilization into electron-rich quinone groups. (**d.3**) Chromium speciation within the UiO-66 framework after the chemical reduction of Cr(VI) to Cr(III), and their coordination to electron-rich quinone groups derived from hydroxyl functions. Possible chromium speciation without Cr(V) transient species within UiO-66 framework containing hydroxyl groups. Reproduced with permission from reference [73].

**Figure 17 nanomaterials-12-04263-f017:**
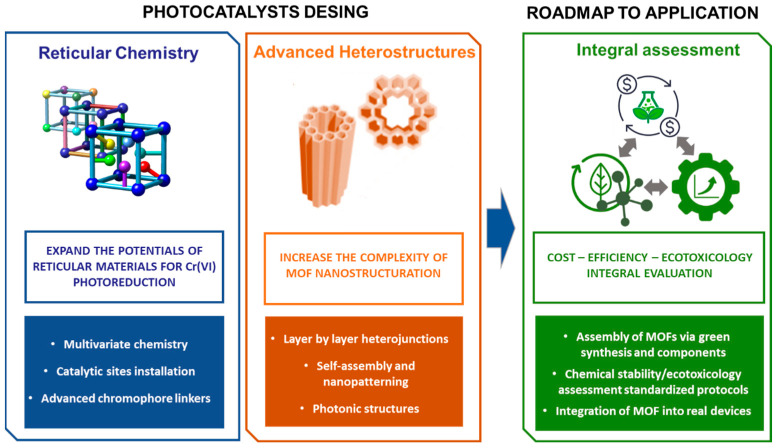
Perspectives to improve the translational success of MOF photocatalysts for chromium photoreduction considering the possible advances from the material design and application requirements points of view.

**Table 2 nanomaterials-12-04263-t002:** Trivalent-metal-based MOF photocatalysts for Cr(VI) to Cr(III) reduction.

Metal	MOFs	pH	Light Source *	[Cr (VI)]_0_	Loading (g/L)	Photo-Oxidation Efficiency			Ref.
						Efficiency (%)		Time (min)	
**Fe**	MIL-53	4	Vis.	20	1	100		40	[223]
	MIL-88B-NH_2_	2	Vis.	8	0.5	100		45	[224]
	MIL-53-NH_2_					15		60	
	MIL-101-NH_2_					100		60	
	MIL-100/HPMo 5%	4	Vis.	20	1	100		8	[225]
	MIL-53/rGO	4	Vis.	20	1	100		80	[226]
	MIL-100/Au 1%	4	Vis.	20	1			20	[227]
	MIL-100/Pd 1%					100		16	
	MIL-100/Pt 1%							8	
	MIL-68/AgBr 30%/Ag 1.5%	4	Vis.	20	0.25	99.9		6	[228]
	MIL-88B-NH_2_/Ag/AgCl	2	-	20	0.5	85.7		45	[229]
	MIL 53/g-C_3_N_4_ 3%	2–3	Vis.	10	0.4	100		180	[230]
	MIL-101-NH_2_ 10%/g-C_3_N_4_	2–3	Vis.	10	0.5	76.0		60	[231]
	MIL-101-NH_2_/g-C_3_N_4_	7	SL	20	1	66		90	[232]
		2				91			
	MIL-68	3	Vis.	20	0.25	100		5	[233]
	MIL-53/WO_3_	2.5	SL	45	1	94		240	[234]
	MIL-100/WO_3_ 80 wt.%/120	2	LED-Vis	5	0.25	100		60	[235]
	NH_2_-MIL 88B/TiO_2_	7	Vis.	10	0.5	98.6		35	[201]
	MIL-53/Bi_12_O_17_C_l2_ 100 mg	2	WL	10	0.5	99.2		120	[236]
	MIL-100/Bi_12_O_17_C_l2_ 200 mg	2	WL	10	0.5	99.3		120	[237]
	MIL-100/PANI 9%	2	WL	10	0.25	100		90	[238]
	Fe-MOF/MoS_2_ 1.5%	2	Vis.	20	1	98.8		60	[215]
	MIL-53	4	Vis.	20	0.5	51		30	[216]
	MIL-53/CQDs/2% Au					100		20	
	MIL-53/CQDs/2% Ag					-		-	
	MIL-53/CQDs/2% Pd					80		30	
	MIL-101-NH_2_/Sand-Cl (50%)	2	Vis. (1000 W)	20	1.0	98.8		60	[239]
	MIL-101-NH_2_/Al_2_O_3_	2	SL	5	0.3	100		8	[240]
	STA-12-Mn-Fe	2	SL	20	0.25	100		30	[241]
	MIL-125-NH_2_/BiO	2	Vis.	40	1	100		120	[242]
**Cr**	MIL-101/Pt	NR	Vis.	NR	NR	100		40	[243]
	MIL-101/Pd							240	
	MIL-101/Pd-Cu	NR	Vis.	NR	NR	100		30	[211]
**In**	MIL-68	2	Vis.	20	1	97		180	[244]
	MIL-68-NH_2_/In_0.4_Fe_0.6_	2	Vis.	20	0.4	99		120	[245]

* Vis. = visible light, SL = sun light, WL = white light.

**Table 3 nanomaterials-12-04263-t003:** Tetravalent-metal-based MOF photocatalysts for Cr(VI) to Cr(III) reduction.

Metal	MOFs	pH	Illumination Source	[Cr (VI)]_0_ (ppms)	Photocatalyst Loading (g/L)	Photo-Oxidation Efficiency		Ref.
						Removal Percentage (%)	Time (min)	
**Zr**	UIO-66-NH_2_	2	Vis.	10	0.5	97	80	[173]
	UiO-66	2	UV/Vis.	10	0.5	35	170	[255]
	UiO-66-NH_2_					100	100	
	UiO-66-NO_2_					12	170	
	UiO-66-Br					22	170	
	UiO-66-NH_2_/rGO	2	Visible	10	0.5	100	100	[256]
	UiO-66/g-C_3_N_4_	2	Visible	10	0.5	99	40	[257]
	UiO-66(OH)_2_/H_2_BDC-(OH)_2_ 20%	2	UV-LED	10	0.4	100	40	[258]
	UiO-66-NH_2_-100/PTCDA-10	2	LED-Visible	10	0.375	100	100	[259]
	UiO-66/BiOBr/Cotton fibers	2.5	Visible	5	2	99	80	[260]
	UiO-66-NH_2−_def	2	Visible	5	0.35	100	100	[46]
	UiO-66-NH_2_/Zr/Hf/ -Al_2_O_3_ membrane	2	Visible	5	-	98	120	[261]
**Ti**	MIL-125/NH_2_	2.1				80	60	[222]
	MIL-125/MoS_2_	6	Visible	48	0.4	20	70	[242]
	MIL-125/Ag_2_S					38		
	MIL-125/CdS					40		
	MIL-125/CuS					60		
	MIL-125-NH_2_/NTU-9	3	Visible	10	1	100	90	[213]
		5				70		
		8				80		
	TiO_2_/MIL-125/core shell	2	Visible	5	0.3	100	60	[243]
	NH_2_-MIL-125/BiOI	2	Visible	40	1	100	120	[211]

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
