# Peer review of "A State-of-the-Art of Metal-Organic Frameworks for Chromium Photoreduction vs. Photocatalytic Water Remediation"

_nanomaterials, 2022, doi:10.3390/nano12234263_

Round 1

Reviewer 1 Report

In this review, Luis et al. summarized the MOFs-based materials and future challenges for the photoreduction of Cr(VI). The removal of Cr(VI) is quite important, and MOFs can be one of the best platforms for this application. This review is well-organized and very comprehensive. Below I suggest some changes and further discussions increase the quality of the review:

1.        Line 57, Page 2, an energy higher than its optical band gap is also suitable. The corresponding sentence should be revised.

2.        Figures, like 10, 13, etc., are not clear.

3.        Most of the Reference numbers are following the punctuations, but few are before them. For example, Line 202, Line 204, etc.

4.        Sacrificial agent, quantum efficiency, space time yield, figure of merit, etc. should better be added in Table 2 (Reference, Coord. Chem. Rev. 2021, 447, 214148).

5.        The summary for the part on the chemical stability of MOFs is not very comprehensive. HSAB principle should better be introduced. References (Adv. Mater. 2018, 30, 1704303; J. Am. Chem. Soc. 2017, 139, 211-217; Sol. RRL 2020, 4, 1900547) can strengthen this aspect.

6.        In the MTV MOF for the photoreduction part, Reference Appl. Catal. B: Environ. 2019, 253, 323-330. is very important, and should better be highlighted. In the Zn MOF, Reference (Ind. Eng. Chem. Res. 2020, 59, 8538) is also important.

7.        Captions for Figures 5a, b, and c should be given. Captions for Figures 8e, and 8f are missing.

8.        For bandgap information, XPS or UPS techniques should also be discussed.

9.        The unit “cm-1” is very suitable, and “nm” is better.

10.     In the Future perspectives part, MOF shaping or filming technologies should be considered. Machine learning can help to screen materials, which is also recommended.

11.     Some typo errors should be carefully checked, for example,

Line 184, an extra “,”.

Line 317, “Metal-Organic Frameworks” should be changed to “MOFs”.

Line 332, Eq. 2 is not complemented.

In line 437, an “.” is missing.

Line 596 has some problems.

Author Response

In this review, Fernandez de Luis et al. summarized the MOFs-based materials and future challenges for the photoreduction of Cr(VI). The removal of Cr(VI) is quite important, and MOFs can be one of the best platforms for this application. This review is well-organized and very comprehensive. Below I suggest some changes and further discussions increase the quality of the review:

First thanks to the referee for the comments. We have tried to implement the manuscript taking into account the mentioned points. You can find a point by point response to your comments/questions below.

  1. Line 57, Page 2, an energy higher than its optical band gap is also suitable. The corresponding sentence should be revised.

We have modified the sentence.

  1. Figures, like 10, 13, etc., are not clear.

After checking different possibilities to make clearer the figures, we have decided to eliminate them from the review. We think that now the manuscript is clearer, and the Figure stile all along the document is more harmonized.

  1. Most of the Reference numbers are following the punctuations, but few are before them. For example, Line 202, Line 204, etc.

This issue has been corrected and the text homogenized.

  1. Sacrificial agent, quantum efficiency, space time yield, figure of merit, etc. should better be added in Table 2 (Reference,  Chem. Rev. 2021,447, 214148).

Thanks to the referee for the suggestion. To be honest, we initiate to assemble this reviews time ago, and unfortunately, we launch it at the same time that other reviews (as the one mentioned by you), were published. The materials reported in both reviews are coincident, as the tables that summarize their performance are. Although, we still believe that our work presents several sections that make the difference with the works that have been already published in this topic.

It would be quite “easy” for us to transfer this data from the Coord. Chem. Rev. review to ours, but we believe that this would not add to much novelty to our work, and above all, that this would not be very fair.

In any case, if the referee think that this update is fully necessary, we will perform in the next round of implementation of the work.

  1. The summary for the part on the chemical stability of MOFs is not very comprehensive. HSAB principle should better be introduced. References ( Mater. 2018,30, 1704303; J. Am. Chem. Soc. 2017, 139, 211-217; Sol. RRL 2020, 4, 1900547) can strengthen this aspect.

We have tried to clarify this section of the review. The references mentioned by the reviewer have been as well included in the manuscript.

  1. In the MTV MOF for the photoreduction part, Reference  Catal. B: Environ. 2019,253, 323-330. is very important, and should better be highlighted. In the Zn MOF, Reference (Ind. Eng. Chem. Res. 2020, 59, 8538) is also important.

We have tried to highlight the seminar review of MTV-MOFs for chromium photoreduction developed by Wang et. al. at the beginning of this section of the review.

The information from the Zn-MOF has been as well included as well within the table 2

  1. Captions for Figures 5a, b, and c should be given. Captions for Figures 8e, and 8f are missing.

We have updated the description of both captions.

  1. For bandgap information, XPS or UPS techniques should also be discussed.

The information has been included within the methodology section of the manuscript.

  1. The unit “cm-1” is very suitable, and “nm” is better.

We have mentioned the units as well in nm in the text.

In the Future perspectives part, MOF shaping or filming technologies should be considered. Machine learning can help to screen materials, which is also recommended.

This is really a good point from the referee. We have incorporated a brief discussion regarding these points in the closing section of the work.

  1. Some typo errors should be carefully checked, for example,

Line 184, an extra “,”.

Line 317, “Metal-Organic Frameworks” should be changed to “MOFs”.

Line 332, Eq. 2 is not complemented.

In line 437, an “.” is missing.

Line 596 has some problems.

We have revisited the review to correct typos.

Reviewer 2 Report

Manuscript Nanomaterials-2013691 (authored by Andreina García et al.)

presents a well-prepared and well-structured review,

the content of which fully corresponds to its Title and Abstract.

I recommend accept it for publication in Nanomaterials after very minor (technical) revision.

The authors should carefully inspect their MS for any misprints and inconsistences.

For example, Figures 16 and 17 are taken from the ref. [73], but not from [66] 

(as indicated in the Legends). By the way, add DOI for [73].

Legends for all non-original Figures have contain 

the references to original papers.

Author Response

I recommend accept it for publication in Nanomaterials after very minor (technical) revision.

The authors should carefully inspect their MS for any misprints and inconsistences.

For example, Figures 16 and 17 are taken from the ref. [73], but not from [66] 

(as indicated in the Legends). By the way, add DOI for [73].

Legends for all non-original Figures have contain the references to original papers.

First, thanks to the referee to review our work. We have solved all these minor errors-mistakes in the revisited manuscript.

Reviewer 3 Report

In this manuscript, the authors discussed and analyzed the state-of-the-art of MOFs for Cr(VI) detoxification and contextualizing it to the most recent advances and strategies of MOFs for photocatalysis purposes. In this minireview, the authors summarized the specific experimental techniques employed to characterize MOF photocatalysts, the key-characteristics of MOFs for Cr(VI) photoreduction, and finally an outlook and perspective section in order to identify future trends. The manuscript is well organized and the synthesis, photodegradation properties and related mechanism of MOFs are systemically summarized. I recommend the manuscript to be published after minor revisions of the language.

Author Response

In this manuscript, the authors discussed and analyzed the state-of-the-art of MOFs for Cr(VI) detoxification and contextualizing it to the most recent advances and strategies of MOFs for photocatalysis purposes.

In this minireview, the authors summarized the specific experimental techniques employed to characterize MOF photocatalysts, the key-characteristics of MOFs for Cr(VI) photoreduction, and finally an outlook and perspective section in order to identify future trends. The manuscript is well organized and the synthesis, photodegradation properties and related mechanism of MOFs are systemically summarized. I recommend the manuscript to be published after minor revisions of the language.

Many thanks to the referee for the comments of our work. We have revisited the manuscript and improve a little again the writing. Hope now the review is more appealing for the reader.

Reviewer 4 Report

The "mini-review" provides general overview of the important topic of MOFs usage for Cr(IV) reduction. As for many review studies there are always relatively similar review reports which present the same aspects of the field. In this context current review is not unique, nevertheless authors did a good job in making this review fluent and easy to understand for the readers outside of the field. The review provides information on all key components of the field including a decent description of key methods which are used to assess the stability of MOFs photocatalysts and some insights for the future developments. By reviewing various sub-topics authors generally provide up-to-date references, but usually do not get into the specific details or findings of the selected studies. For some this might look as a drawback, but in my view considering the provided width of the review, the depth of each sub-topic is adequate. On the other hand, it had to be noticed that in current state the manuscript still has a lot of technical "issues" (I provide most of my noticed issues bellow) which has to corrected before it could be published.

Summing up I find the presented manuscript of good quality and would recommend it for the publishing after large scale review and correction of "technical issues". 

Noticed issues:

Figures and tables should be placed closer to their first mentioning in the text. Preferably at the same page of the first usage.

Authors should review the usage of numbers in formulas and other indexes. Throughout the manuscript a lot of them are placed as normal font, but should be provided as subscripts.

Information presentation in tables should be reviewed and corrected for more better clearance. If some of the lines share the information from the same references, or have the same experimental conditions, this should be clearly marked by using corresponding centring, line marking or other measures.

line 168 wrong figure reference

lines 241-244 non-uniform numbering style

lines 270-300 The section describing methods to estimate chemical and hydrolic stability of MOFs in water needs to be reviewed and corrected. There is incorrect referencing to the Figure 5 and/or its corresponding sub-figures. Also, the caption of Figure 5 is not informative. I would suggest to expand it with some short remarks on what information specific methods provide and what are the differences between the coloured curves. Presumably, the provided curves are just illustrations for some model/"dummy" cases, but this should be stated more clearly. Similarly, the same clarification is suggested and for other Figures with unmarked curves.

lines 484-487 wrong caption for figure 7 is provided (the copy from Figure 6).

lines 559-564 wrong caption for figure 8 is provided (the copy from Figure 6).

lines 596-598 part of the sentence was omitted and needs to be "restored".

line 632 table number and caption should be corrected.

lines 720-724 figure should be placed at the section of divalent metal-MOFs, but it is never mentioned in the corresponding paragraphs of the text. Later in the section for trivalent metal-MOFS the figure 9 (line 689) is referenced incorrectly. In fact, at line 689 the reference should point to Figure 10, and all following figure captions should be reviewed accordingly.

line 741 - Figure 10 quality is inappropriate.

lines 798-802 Expressions in Figure caption should be reviewed.

lines 771-778 there are several observations provided, but references. This should be corrected

line 819 - is it pH<3 or pH<4? also it seems, that the significance of hole trapping agents was not reviewed in the text, though at the summary section it is stated as one of the key factors. 

lines 875-881 authors provide the full text for MIL, UiO, but not for DUT. Also, it can be noted, that if one would like to provide full texts for the abbreviations, this should be done at its first mentioning, not in the third part of the manuscript.

lines 853, 859 wrong figure reference

line 959 the captions in Figure are very small and not comfortable for reading. Accordingly authors are suggested to enlarge the Figure or to find other way to improve readability.

lines 982-987 caption should be reviewed.

lines 1014-1016 Figure reference should be reviewed

line 1037 Who are "they"?

lines 1046-1049 figure captions one-to-one repeat the text on the Figure. Authors are suggested either to crop the top captions, or to reformulate the main Figure caption.

line 1059 bad figure referencing

reference list formatting and uniformity should be reviewed.

Some other parts of the manuscript also require revisions, for example lines: 106-107, 181-182, 184, 315, 317, 321, 324, 383, 387, 426, 429-431, 479-480, 537-538, 544, 577, 640, 689, 704-707, 764, 809, 825, 826, 852, 859, 1054,

Author Response

REFEREE-4

The "mini-review" provides general overview of the important topic of MOFs usage for Cr(IV) reduction. As for many review studies there are always relatively similar review reports which present the same aspects of the field.

In this context current review is not unique, nevertheless authors did a good job in making this review fluent and easy to understand for the readers outside of the field. The review provides information on all key components of the field including a decent description of key methods which are used to assess the stability of MOFs photocatalysts and some insights for the future developments. By reviewing various sub-topics authors generally provide up-to-date references, but usually do not get into the specific details or findings of the selected studies.

For some this might look as a drawback, but in my view considering the provided width of the review, the depth of each sub-topic is adequate. On the other hand, it had to be noticed that in current state the manuscript still has a lot of technical "issues" (I provide most of my noticed issues bellow) which has to corrected before it could be published.

Summing up I find the presented manuscript of good quality and would recommend it for the publishing after large scale review and correction of "technical issues".

We would like to thanks the referee for the great effort developed to evaluate our work. We really appreciate all the comments/points/questions. We have summarized below the response to the reported comments/technical issues. Hopefully the work has been published a little bit more after addressing the weakness identified by the referee.

Figures and tables should be placed closer to their first mentioning in the text. Preferably at the same page of the first usage.

We have re-placed the Figures and the tables to accomplish this requirement. In the specific case of the tables, its more difficult to achieve this because of their size, but we have tried to do so.

Authors should review the usage of numbers in formulas and other indexes. Throughout the manuscript a lot of them are placed as normal font, but should be provided as subscripts.

The subscripts of the numbers all along the manuscript have been reviewed.

Information presentation in tables should be reviewed and corrected for more better clearance. If some of the lines share the information from the same references, or have the same experimental conditions, this should be clearly marked by using corresponding centring, line marking or other measures.

A differentiation between the materials corresponding to the same reference, or applied under the same photocatalysis conditions have been applied in the tables. In addition, some minor mistakes have been corrected as well.

line 168 wrong figure reference

lines 241-244 non-uniform numbering style

Corrected

lines 270-300 The section describing methods to estimate chemical and hydrolic stability of MOFs in water needs to be reviewed and corrected. There is incorrect referencing to the Figure 5 and/or its corresponding sub-figures. Also, the caption of Figure 5 is not informative. I would suggest to expand it with some short remarks on what information specific methods provide and what are the differences between the coloured curves. Presumably, the provided curves are just illustrations for some model/"dummy" cases, but this should be stated more clearly. Similarly, the same clarification is suggested and for other Figures with unmarked curves.

We have updated the caption of the figures and include more specific descriptions into this section of the manuscript.

lines 484-487 wrong caption for figure 7 is provided (the copy from Figure 6).

The caption has been corrected

lines 559-564 wrong caption for figure 8 is provided (the copy from Figure 6).

The caption has been corrected

lines 596-598 part of the sentence was omitted and needs to be "restored".

Done

line 632 table number and caption should be corrected.

The table number and the caption have been corrected

lines 720-724 figure should be placed at the section of divalent metal-MOFs, but it is never mentioned in the corresponding paragraphs of the text.

This issue has been corrected.

Later in the section for trivalent metal-MOFS the figure 9 (line 689) is referenced incorrectly. In fact, at line 689 the reference should point to Figure 10, and all following figure captions should be reviewed accordingly.

line 741 - Figure 10 quality is inappropriate.

These figures have been removed to make the review more consistent and clearer.

lines 798-802 Expressions in Figure caption should be reviewed.

Done

lines 771-778 there are several observations provided, but references. This should be corrected

Done

line 819 - is it pH<3 or pH<4? also it seems, that the significance of hole trapping agents was not reviewed in the text, though at the summary section it is stated as one of the key factors. 

Done

lines 875-881 authors provide the full text for MIL, UiO, but not for DUT. Also, it can be noted, that if one would like to provide full texts for the abbreviations, this should be done at its first mentioning, not in the third part of the manuscript.

We have preferred to eliminate this information. 

lines 853, 859 wrong figure reference

Corrected

line 959 the captions in Figure are very small and not comfortable for reading. Accordingly authors are suggested to enlarge the Figure or to find other way to improve readability.

Solved.

lines 982-987 caption should be reviewed.

We have shortened a little the caption and modify the

lines 1014-1016 Figure reference should be reviewed

Done

line 1037 Who are "they"?

We have detailed the reference.

lines 1046-1049 figure captions one-to-one repeat the text on the Figure. Authors are suggested either to crop the top captions, or to reformulate the main Figure caption.

We have shortened the caption.

line 1059 bad figure referencing

Solved.

reference list formatting and uniformity should be reviewed.

Done

Some other parts of the manuscript also require revisions, for example lines: 106-107, 181-182, 184, 315, 317, 321, 324, 383, 387, 426, 429-431, 479-480, 537-538, 544, 577, 640, 689, 704-707, 764, 809, 825, 826, 852, 859, 1054,

We have revisited these sections together with other ones that we have found in a deep re-checking of the manuscript.

Round 2

Reviewer 4 Report

Appearingly, authors have addressed most of the issues which there identified during the first revision. Therefore, my suggestion is to accept the manuscript in its present form.